# A novel role for the peptidyl-prolyl *cis-trans* isomerase Cyclophilin A in DNA-repair following replication fork stalling via the MRE11-RAD50-NBS1 complex

Marisa Bedir [ID], Emily Outwin [ID], Rita Colnaghi, Lydia Bassett [ID], Iga Abramowicz [ID] & Mark O'Driscoll [ID] [✉]

## Abstract

**Cyclosporin A (CsA) induces DNA double-strand breaks in LIG4 syndrome fibroblasts, specifically upon transit through S-phase. The basis underlying this has not been described. CsA-induced genomic instability may reflect a direct role of Cyclophilin A (CYPA) in DNA repair. CYPA is a peptidyl-prolyl *cis-trans* isomerase (PPI). CsA inhibits the PPI activity of CYPA. Using an integrated approach involving CRISPR/Cas9-engineering, siRNA, BioID, co-immunoprecipitation, pathway-specific DNA repair investigations as well as protein expression interaction analysis, we describe novel impacts of CYPA loss and inhibition on DNA repair. We characterise a direct CYPA interaction with the NBS1 component of the MRE11-RAD50-NBS1 complex, providing evidence that CYPA influences DNA repair at the level of DNA end resection. We define a set of genetic vulnerabilities associated with CYPA loss and inhibition, identifying DNA replication fork protection as an important determinant of viability. We explore examples of how CYPA inhibition may be exploited to selectively kill cancers sharing characteristic genomic instability profiles, including MYCN-driven Neuroblastoma, Multiple Myeloma and Chronic Myelogenous Leukaemia. These findings propose a repurposing strategy for Cyclophilin inhibitors.**

**Keywords** Cyclophilin A; DNA End Resection; MRE11-RAD50-NBS1; Cyclosporin A; Repurposing Cyclophilin Inhibitors
**Subject Category** DNA Replication, Recombination & Repair

## Introduction

Cyclosporin A (CsA) is a cyclic undecapepetide originally isolated from the fungus *Tolypocladium inflatum* (Fischer and Malesevic, 2013; Fischer et al, 1989; Rüegger et al, 1976). Although CsA does not bind DNA directly, we and others have found that it causes DNA breakage and genome instability under certain circumstances (IARC, 1990; O'Driscoll and Jeggo, 2008; Oztürk et al, 2008; Palanduz et al, 1999; Yuzawa et al, 1986). Importantly, the molecular basis underlying this has not yet been described. We hypothesised that CsA's impact on genome integrity could be via Cyclophilin A (CYPA), an abundant and principal CsA interacting target (Handschumacher et al, 1984; Harding and Handschumacher, 1988; Kallen et al, 1998; Takahashi Hayano and Suzuki, 1989; Zydowsky et al, 1992). Our reasoning was centred on the fact that CYPA possesses a physiologically significant intrinsic enzymatic activity as a peptidyl-prolyl *cis-trans* isomerase (PPI) (Davis et al, 2010; Rajiv and Davis, 2018; Wang and Heitman, 2005). This PPI activity is inhibited by CsA (Handschumacher et al, 1984; Harding and Handschumacher, 1988; Kallen et al, 1998; Takahashi Hayano and Suzuki, 1989; Zydowsky et al, 1992). Other classes of PPIs, such as the well-known Parvulin family member PIN1 and certain FKBPs (FK506-binding proteins), directly influence DNA repair and the DNA damage response (Dilworth et al, 2020; Hilton et al, 2015; Steger et al, 2013; Wang et al, 2019). CYPA's PPI function is implicated in supervising optimal protein folding, in regulating local structural transitions influencing protein function by essentially acting as a molecular rheostat, and as a 'holdase' in controlling liquid-liquid phase separation (LLPS) of certain intrinsically disordered proteins (IDPs) and proteins with defined regions of disorder (Adams et al, 2015; Andreotti, 2003; Babu et al, 2022; Favretto et al, 2020a; Favretto et al, 2020b; Hill et al, 2022; Lu et al, 2007; Schmidpeter and Schmid, 2015; Wedemeyer Welker and Scheraga, 2002a; Xia and Levy, 2014; Zhang et al, 2013; Zheng et al, 2008).

As our original observation was that CsA-induced DSBs in LIG4 syndrome patient fibroblasts that had specifically traversed S-phase, we sought to examine how CYPA PPI function impacts upon DNA repair by focusing on mechanisms active during S-phase (O'Driscoll and Jeggo, 2008). DNA replication forks that stall upon encountering endogenous lesions such as single-stranded breaks, covalent DNA-adducts and DNA-protein crosslinks, must be stabilised and protected from uncontrolled nuclease activity

Human DNA Damage Response Disorders Group, Genome Damage & Stability Centre, University of Sussex, Brighton BN1 9RQ, UK. ✉E-mail: m.o-driscoll@sussex.ac.uk

(Gaillard Garcia-Muse and Aguilera, 2015; Pasero and Vindigni, 2017). Such aberrantly unrestrained activity could result in loss of genetic information and the initiation of genomic rearrangements (Al-Zain and Symington, 2021; Berti Cortez and Lopes, 2020; Kramara Osia and Malkova, 2018; Pasero and Vindigni, 2017). Consistent with this, complex DNA rearrangements have been observed following treatment with CsA (O'Driscoll and Jeggo, 2008; Ozturk et al, 2008; Palanduz et al, 1999; Yuzawa et al, 1986). Conversely, accurate homologous recombination repair (HRR) during S-phase is initiated by the coordinate and highly regulated actions of these same nucleases through controlled resection initiation orchestrated by CtIP (Andres and Williams, 2017; Cejka and Symington, 2021; Makharashvili and Paull, 2015; Mozaffari Pagliarulo and Sartori, 2021). CtIP is activated and restrained through a multitude of protein-protein interactions which are regulated via ATM/R and CDK-dependent phosphorylations on CtIP (Andres and Williams, 2017; Cejka and Symington, 2021; Makharashvili and Paull, 2015; Mozaffari Pagliarulo and Sartori, 2021). One of the most important interacting partners of CtIP is the MRE11-RAD50-NBS1 (MRN) complex (Paull, 2018; Syed and Tainer, 2018). MRN plays a lead role in resection initiation and propagation in coordination with other nucleases at DSBs and stalled DNA replication forks, thereby directing these structures to the most appropriate major DNA repair pathway, including HRR, single-strand annealing (SSA) or NHEJ (Britton et al, 2020; Cejka and Symington, 2021; Chanut et al, 2016; Reginato and Cejka, 2020; Shibata Jeggo and Löbrich, 2018).

In this study, we show that CYPA co-immunoprecipitates with several members of the end resection machinery and describe a novel direct interaction between CYPA and an MRN component. We show that when cells are compromised through CYPA loss or inhibition, MRN function is disrupted following DNA replication fork stalling, severely limiting end resection and consequently markedly impairing resection-mediated repair pathways, including HRR and SSA. We also identify a set of genetic vulnerabilities associated with CYPA loss and inhibition, exposed by this altered DNA-R landscape, before exploring examples of how CYPA inhibition may have clinical efficacy in a defined set of cancers with a shared genome instability profile involving aberrantly elevated/addiction to HRR.

Collectively, our findings represent completely new biological insight into CYPA; how its PPI activity influences DNA repair, new vulnerabilities following CYPA loss and inhibition, as well how CYPA inhibition selectively induces cytotoxicity in specific cancer cell lines. The latter has implications for the repurposing/ repositioning of CYPA inhibitors; specifically, the expansive range of non-immunosuppressive CsA analogues (NIAs).

## Results

### Combining CYPA loss or inhibition with impaired NHEJ

CsA can induce DSBs and complex rearrangements, including sister chromatid exchanges (SCEs) and fusions (IARC, 1990; O'Driscoll and Jeggo, 2008; Ozturk et al, 2008; Palanduz et al, 1999; Yuzawa et al, 1986) (Fig. 1A,B). It is likely that this reflects the sensitivity of human pre-B LIG4−/− cells we originally reported (O'Driscoll and Jeggo, 2008). Importantly, whilst CYPA appears to

be CsA's main interactor, CsA can bind and inhibit a variety of other Cyclophilin family members (Davis et al, 2010; Hu et al, 2014). To study CYPA specifically, we created model U2OS lines including a knockout (KO) for *PPIA*/CYPA using CRISPR/Cas9 and an isogenic KO variant reconstituted with a catalytically 'isomerase-dead' CYPA engineered to be p.R55A (R55A) (Zydowsky et al, 1992) (Fig. 1C). To investigate whether loss of the CsA binding target CYPA specifically underlies the sensitivity observed under impaired LIG4 activity we undertook siRNA of LIG4 in this engineered panel of U2OS cells. We found decreased clonogenic survival upon siLIG4 in CYPA-KO cells and in KO cells reconstituted with a catalytically dead CYPA (R55A) (Fig. 1D,E). The latter indicates that inhibition of the PPI activity of CYPA is synthetic lethal with reduced LIG4 expression. Interestingly, we also found that Ku80-defective CHO cells (xrs-6) were markedly sensitive to killing by CsA (Fig. 1F).

### CYPA and DNA replication

During routine culturing of our CYPA-engineered isogenic cell line panel we noticed that both the CYPA-KO and CYPA-R55A lines appeared to cycle slower compared to the scrambled (Scrm)-control line. To investigate this at a more fundamental cell cycle phase level, we performed EdU pulse with and without a prolonged mitotic trap using nocodazole (Noc). Decreased DNA replication was demonstrated in the CYPA-KO and CYPA-R55A lines following a short EdU pulse (Fig. 2A, EdU panels). Using EdU and nocodazole produced a strong lagging tail of EdU-positive cells with S-phase DNA content after 24 h in CYPA-KO and CYPA-R55A, in stark contrast to those of the scrambled (Scrm)-control line, where all EdU positive cells had attained a 4 N DNA content indicative of S-phase transit into mitosis (Fig. 2A, EdU+Noc panels). Our results indicate that CYPA-KO and CYPA-R55A display a delayed S phase transit.

Consistent with this, DNA fibre combing showed reduced DNA replication fork speed in the CYPA-KO and CYPA-R55A cells compared to the scrambled (Scrm)-control and KO reconstituted with wild-type (WT) CYPA (Fig. 2B; Appendix Fig. S1A). In addition, we found elevated levels of fork stalling following treatment with hydroxyurea (HU) in the CYPA-KO cells compared to control lines (Fig. 2C). ATR-dependent CHK1 phosphorylation was comparable between the scrambled (Scrm)-control, CYPA-KO and CYPA-R55A lines (Appendix Fig. S1B).

### CYPA and DSB repair pathway function

To directly assess the impact of CYPA silencing on the principal DSB repair pathways we undertook analysis using the DR-GFP-reporter platform integrated into U2OS cells. A novel variation of the system we employed was the use of a doxycycline-inducible conditional I SceI via the *Sleeping Beauty* transposon system, which enabled consistent and high-level expression of the restriction enzyme and consequently high levels of GFP signal (~15% in control (Ctrl) lines; *c/o Dr. Owen Wells*). We found that siCYPA results in a reproducible ~20% reduction in both NHEJ and SSA, with a more marked reduction of 40–50% in HRR activity in this system (Fig. 3A–C). The large impairment of HRR activity would explain the sensitivity of NHEJ-defective cells we reported previously and expand upon here. We also found that CsA

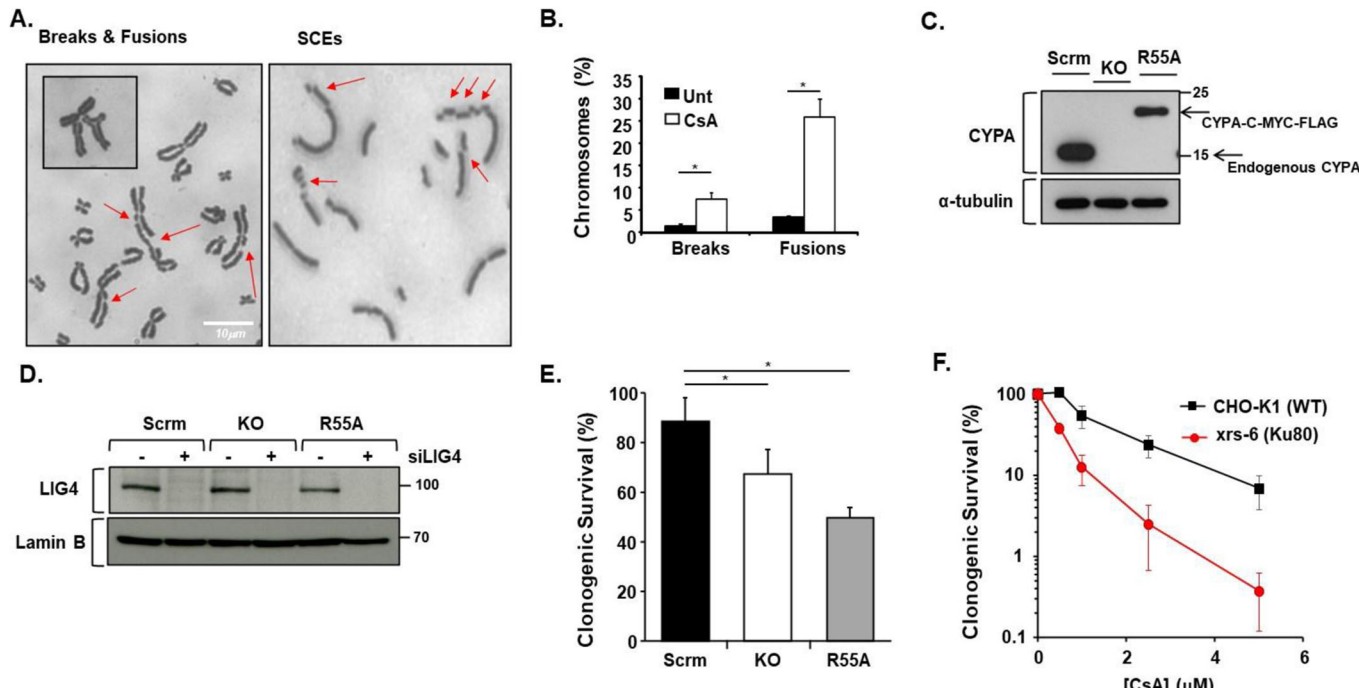

**Figure 1. CYPA inhibition causes chromosomal damage and reduced viability when NHEJ is compromised.**

(A) Images showing the array of diverse chromosome aberrations observed following prolonged treatment of AA8 CHO cells with CsA (5 μM, 24 h). Nocodazole (1 nM, 24 h) was also included to trap mitotic chromosomes. In the lefthand image panel individual breaks and various fusion events are marked with the red arrows. An extreme example of the latter is shown in the inset panel. We also found that CsA (5 μM, 12 h) induced sister chromatid exchanges (SCEs) as indicated by the red arrows in the righthand image panel. Scale bar: 10 μm. (B) Quantification of the breakage and fusion events observed following CsA treatment of these AA8 cells (error bars are mean ± s.d. of 3× independent determinations *CsA-Breaks $P = 0.0046$, *CsA-Fusions $P = 0.0001$, Student's $t$ test). Fusion-type events, perhaps indicative of aberrant DNA repair outcomes were commonly observed events under these conditions. (C) CRISPR/Cas9 knockout of *PPIA*/CYPA and reconstitution in U2OS cells. Scrm denotes scrambled gRNA. KO denotes *PPIA*/CYPA knockout. R55A was stably reconstituted with C-terminally MYC-FLAG-tagged *PPIA*/CYPA engineered to be p.R55A, which is a peptidyl-prolyl *cis-trans* isomerase (PPI) catalytically dead variant (R55A). This line was employed throughout to ascertain whether the PPI activity of CYPA was required for whatever endpoint was being investigated. The upper panel shows CYPA expression. Note the absence of endogenous CYPA in the KO (and R55A), as expected. The lower panel confirms protein loading throughout via α-tubulin expression. (D) Western blot analysis showing effective silencing of LIG4 (siLIG4) in the U2OS isogenic panel of scrambled (Scrm), *PPIA*/CYPA knockout (KO) and CYPA p.R55A PPI-dead (R55A). Lamin B was used to confirm loading across the panel. (E) Clonogenic survival of the U2OS isogenic panel following silencing of LIG4. We found reduced survival following transient siLIG4 in the knockout (KO) and PPI-dead (R55A) cell lines relative to the scrambled (Scrm) control, in the absence of exogenously supplied DNA damage (error bars indicate the mean ± s.d. of $n = 3$ independent determinations *KO $P = 0.0065$, *R55A $P = 0.0014$, Student's $t$ test). (F) Clonogenic survival following treatment with increasing doses of CsA shows that Ku80-defective CHO cells (xrs-6) are markedly sensitive compared to their parental control line (CHO-K1) (error bars indicate the mean ± s.d. of $n = 3$ independent determinations). Source data are available online for this figure.

treatment similarly impairs these distinct repair pathways, which is consistent with the PPI activity of CYPA being required for their optimal function (Fig. 3D). In addition, even greater reductions in the proficiency of repair in all pathways but particularly the resection-driven DSB repair pathways SSA and HRR were found following treatment with NIM811 (*N-methyl-4-isoleucine*-cyclosporine), a non-immunosuppressive CsA analogue which exhibits a much stronger affinity for CYPA compared to CsA (Fu et al, 2014; Lawen et al, 1994) (Fig. 3D). Therefore, we find that CYPA loss and inhibition inhibits DSB repair, particularly resection-driven pathways.

## A BioID-derived CYPA proximity interactome

To obtain a more detailed understanding of how CYPA influences genome stability, we generated a BioID coupled-mass spectroscopy proximity interactome of human CYPA in the presence and absence of DNA replication fork stalling using HU (Roux et al, 2012). We identified over 400 unique 'hits', (Dataset EV1, BioID

Protein list). Their respective genes (Dataset EV2, BioID Gene list) were used to compile a gene ontology (GO) overview using the PANTHER platform, segregating into 37 biological processes (Dataset EV3, BioID GO). The most significantly enriched GO biological processes are summarised in Fig. 4. The enrichment of RNA processes such as transcription, RNA metabolism and splicing are in-keeping with CYPA's known role as a chaperone for newly synthesised proteins and with the association of the Cyclophilin family of PPIs with the spliceosome (Adams et al, 2015; Rajiv and Davis, 2018; Schmidpeter and Schmid, 2015; Wedemeyer Welker and Scheraga, 2002b) (Fig. 4A). Importantly, we also observed significant enrichment of GO biological processes contributing to DNA repair, recombination, and cell cycle regulation, generating an abundant trove of potentially novel physiologically relevant targets of CYPA to pursue (Fig. 4A,B). A summary of the proximity interactions detected between CYPA and targets implicated in the mechanics of cell division, RNA processing and transcription are detailed in Appendix Tables S1 and S2.

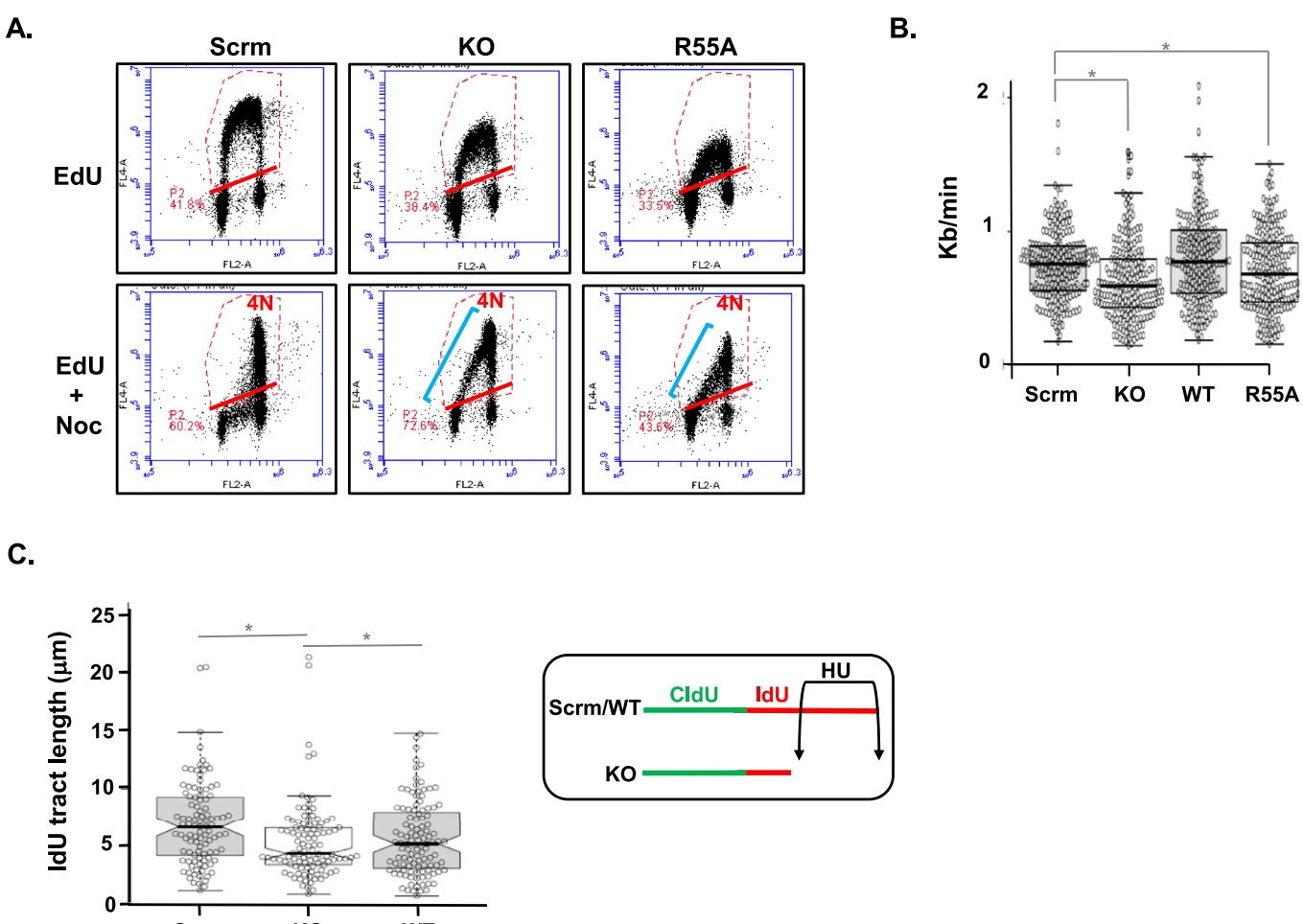

**Figure 2.  CYPA loss and inhibition impairs S phase progression and increases DNA replication fork stalling.**

(A) The upper panels show flow cytometry profiles of scrambled (Scrm), knockout (KO) and PPI-dead (R55A) U2OS cells following pulse labelling (30 min) with EdU to indicate S-phase content. The lower panels show the profiles of these cells following prolonged (24 h) treatment with EdU and nocodazole (Noc), to illustrate the progressing of S-phase cells (EdU +ive) towards a Noc-induced mitotic block (4 N DNA content). The EdU pulse data (upper panels) indicates reduced EdU incorporation in both the KO and R55A compared to the Scrm control, suggesting reduced DNA synthesis under these conditions. In the EdU + Noc treated cells (lower panels), both the KO and R55A lines show a substantial number of cells remaining in S-phase (i.e. with <4 N DNA content) after 24 h (blue line). This is in stark contrast to the scrambled (Scrm) control cell line where all cells have attained a 4 N DNA content indicative of successive progression from S-phase into mitosis. These data show that knockout and expression of a catalytically dead CYPA (i.e., R55A) markedly impair S-phase progression. (B) DNA fibre combing analysis of CldU and IdU labelled DNA replication forks show significantly reduced fork speed following knockout of *PPIA*/CYPA (KO) and expression of isomerase-dead (R55A) cells compared to scrambled (Scrm) control isogenic U2OS cells and KO cells reconstituted with wild-type (WT) *PPIA*/CYPA (*KO $P = 2.59 \times 10^{-5}$, *R55A $P = 0.0487$, Student's *t* test. Scrm $n = 204$, KO $n = 200$, WT $n = 202$ and R55A $n = 200$). The horizontal line within each box represents the median, and the box boundaries are defined by the 25th and 75th percentiles. The whiskers extend to the minimum and maximum values. Median values are Scrm = 0.76, KO = 0.59, WT = 0.77 and R55A = 0.68. (C) Following CldU labelling (20 min) cells were treated with HU (1.5 mM, 2.5 h) in the presence of IdU and DNA fibres combed to examine the extent of DNA replication fork stalling (insert). We found significantly reduced IdU tract lengths in *PPIA*/CYPA knockout U2OS (KO) compared to scrambled (Scrm) control isogenic U2OS cells and KO cells reconstituted with wild-type (WT) *PPIA*/CYPA (*KO $P = 0.0008$, *WT $P = 0.0036$, Student's *t* test. Scrm $n = 101$, KO $n = 105$, WT $n = 111$). The horizontal line within each box represents the median, and the box boundaries are defined by the 25th and 75th percentiles. The whiskers extend to the minimum and maximum values. Median values are Scrm = 6.58, KO = 4.25, WT = 5.09. Source data are available online for this figure.

## CYPA interacting protein partners involved in DNA repair

Figure 5A shows a summary of the proximity interactions detected between CYPA and proteins involved in DNA repair and replication.

Using immunoprecipitation (IP) of endogenous CYPA from HEK293 cells, we sought to validate some of the putative interactors with known roles in genome stability identified by CYPA-BioID. Additionally, to investigate the dependencies of these protein-protein interactions upon the PPI activity of CYPA, we undertook co-IP analyses in extracts from HEK293 cells treated with the CYPA inhibitor CsA. We observed endogenous interaction by co-IP of CYPA with PCNA, 53BP1, CHAMP1 and the ILF2-3 complex (Fig. 5B–E). Interestingly, a direct interaction between CYPA and PCNA has been documented by FAR western analysis, corroborating our approach (Naryzhny and Lee, 2010). Importantly, all these co-IP interactions were sensitive to treatment with CsA, suggesting the PPI activity and/or access to the catalytic active site of CYPA is essential to maintain these interactions (Fig. 5B–E).

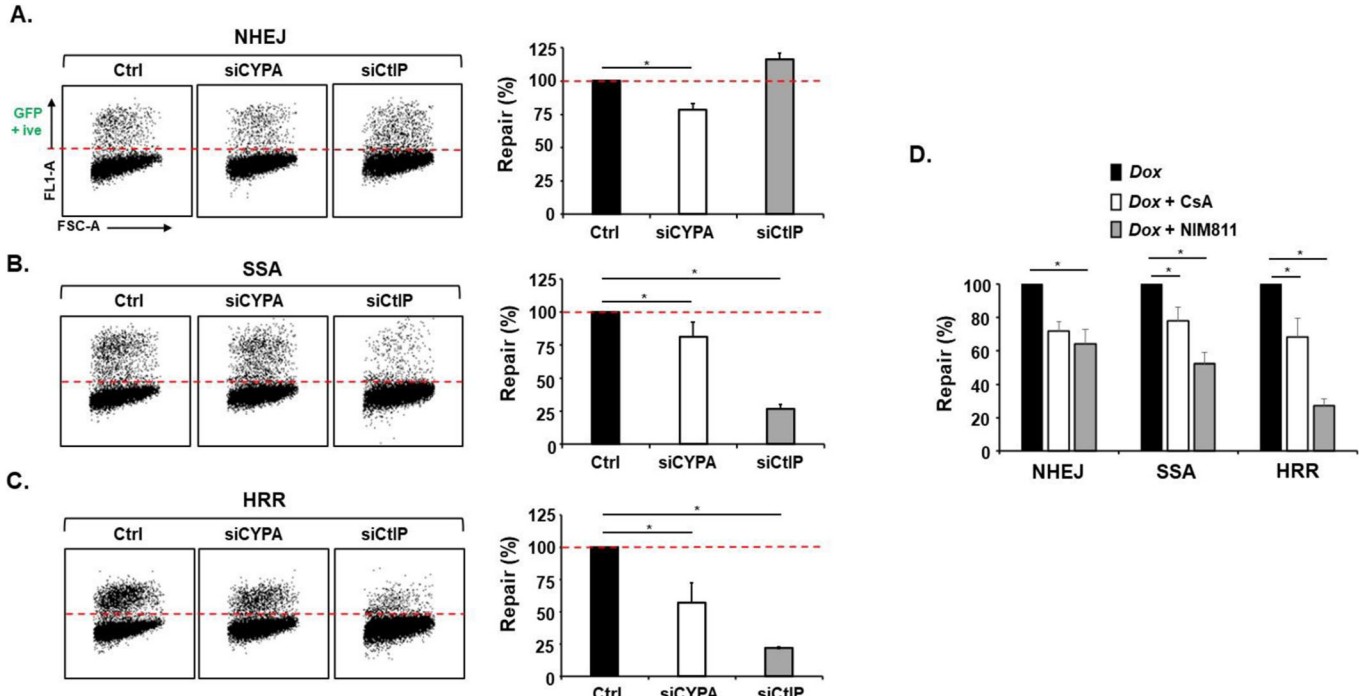

**Figure 3. CYPA loss and inhibition impairs resection mediated homology-directed DNA repair.**

(A) The lefthand panels show individual flow cytometry profiles of a U2OS line containing a GFP-reporter cassette to monitor I SceI-induced non-homologous end-joining (NHEJ) following transfection with control (Ctrl) siRNA, siCYPA and siCtIP. This variation of the standard reporter system employs a doxycycline (Dox) inducible I SceI that results in greater expression and consequently a larger amount of GFP +ive cells indicative of effective DNA repair. Confirmation of the extent of gene silencing in all lines is shown in Appendix Fig. S1C,D. GFP +ive signal indicates efficient NHEJ (above the dotted red line). The bar chart on the righthand side shows the mean ± s.d. of 5× independent replicates (*siCYPA $P = 0.0004$, Student's $t$ test). The dotted red line on the bar chat indicates the normalised NHEJ signal of the control (Ctrl) cell line. Silencing CYPA results in a modest although significant ~25% reduction in NHEJ. As demonstrated previously, siCtIP results in a slight increase in NHEJ. (B) Here, the lefthand panels show individual flow cytometry profiles of the U2OS line containing the GFP-reporter cassette to monitor I SceI-induced single-strand annealing (SSA). When SSA efficiency is normalised to that of the control (Ctrl) cells, similar to our findings with the NHEJ reporter system, the bar chart on the righthand side shows that siCYPA results in a modest but significant ~20% reduction in SSA compared to the control (Ctrl) cells. As expected siCtIP results is a marked reduction of SSA consistent with its role as a master regulator of DSB resection (mean ± s.d. of 5× independent replicates. *siCYPA $P = 0.0004$, *siCtIP $P = 1.62 \times 10^{-7}$, Student's $t$ test). (C) The lefthand panels show individual profiles of the U2OS line containing the GFP-reporter cassette to monitor I SceI-induced homologous recombination repair (HRR). The bar chart on the righthand side shows that siCYPA results in a marked ~50% reduction in HHR compared to control (Ctrl) cells. Again, as expected siCtIP results is a marked reduction in resection-dependent HRR (mean ± s.d. of 5× independent replicates. *siCYPA $P = 2.72 \times 10^{-5}$, *siCtIP $P = 5.48 \times 10^{-11}$, Student's $t$ test). (D) The U2OS reporter lines for each of the main DSB repair pathways (NHEJ, SSA and HRR) where treated with CsA (5 µM every 24 h up to 48 h) or the non-immunosuppressive CsA analogue NIM811 (*N-methyl-4-isoleucine*-cyclosporine. 5 µM every 24 h up to 48 h) in the presence of Dox, and GFP signal quantified and normalised to Dox-induced only repair proficient control cells. CYPA inhibition with CsA results in significantly reduced NHEJ, SSA and HRR. NIM811 treatment produced a similar trend of reduced DNA repair although this was more marked compared to CsA for both SSA and HRR. NIM811 is a more potent inhibitor of CYPA compared to CsA. Collectively, the outcome from using both inhibitors is that inhibition of the PPI function of CYPA significantly reduces DSB repair across all of the principal pathways, including those dependent upon end resection (mean ± s.d. of 5× independent replicates. *NHEJ, CsA $P = 0.0007$, NIM811 $P = 0.0045$, *SSA CsA $P = 0.0003$, NIM811 $P = 0.0069$, *HRR, CsA $P = 0.0005$, NIM811 $P = 0.0011$, Student's $t$ test). Source data are available online for this figure.

The PPI-sensitive nature of these interactions is consistent with CYPA binding to and stabilising the planer transition state between *cis-trans* of the target prolyl peptide bond (Andreotti, 2003; Ladani et al, 2015), and with CYPA being poised to influence the function of its binding partners, effectively acting as an intrinsic molecular switch (Andreotti, 2003; Lu et al, 2007).

53BP1 plays an important role in countering BRCA1-dependent resection, favouring NHEJ (Bouwman et al, 2010; Bunting et al, 2010; Cao et al, 2009; Zimmermann and de Lange, 2014; Zong Ray Chaudhuri and Nussenzweig, 2016). CHAMP1 promotes resection and HRR via interaction with REV7/MAD2L2/FANCV, and together with POGZ counteracts the inhibitory effect of 53BP1 on HRR (Fujita et al, 2022; Li et al, 2022; Sale, 2015). ILF2-3 complex interacts with 53BP1, MDC1, BRCA1, RPA1-3, MCM2-7, the DNA-PK complex, MRE11 and RAD50 (Gupta et al, 2018; Nourreddine et al, 2020; Shamanna et al, 2011; Ting et al, 1998;

Wandrey et al, 2015). In fact, siILF2 was reported to reduce MRN complex recruitment to the β-globin locus and impair both NHEJ and HRR (Karmakar et al, 2010; Marchesini et al, 2017; Shamanna et al, 2011).

A recurring theme amongst these CYPA interactors is that all are involved in end resection, an observation which directed our subsequent research trajectory, and which is consistent with the precise constellation of DNA repair defects we identified regarding HRR and SSA (Fig. 3B–D).

## CYPA and end resection

We employed HU-induced pRPA2 (S4/S8) to examine resection-mediated ssDNA formation under conditions of DNA replication fork stalling. Initially, we found that acute treatment with CsA significantly impaired pRPA2 formation in U2OS suggesting PPI

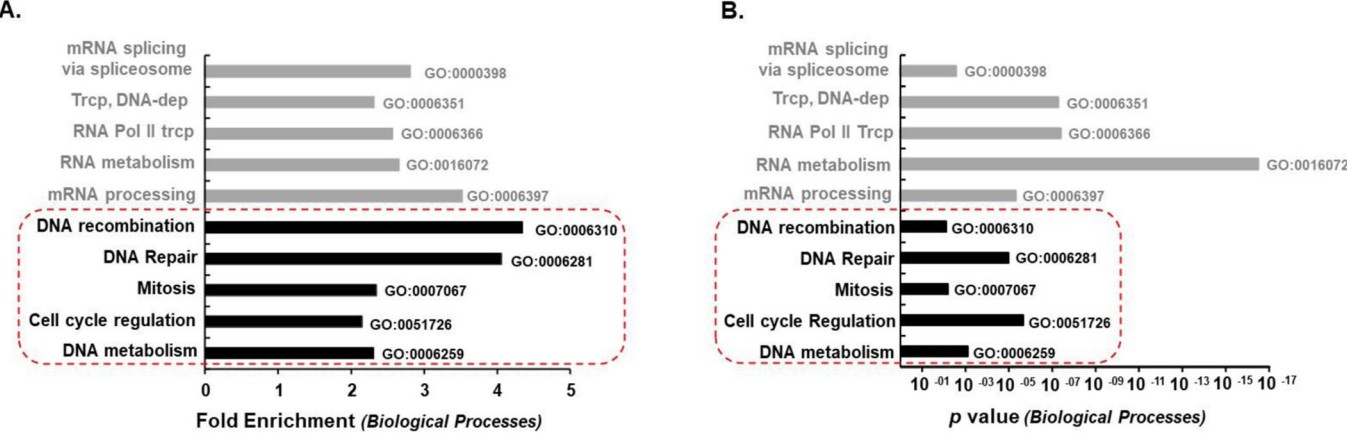

**Figure 4. CYPA BioID reveals putative interactors in pathways that control genome stability and cell cycle.**

(A) A summary of the some of the most enriched CYPA-BioID-derived target genes following gene ontology (GO) analysis using PANTHER and classification according to '*Biological processes*'. The complete list of protein hits is detailed in Dataset EV1 and their corresponding genes in Dataset EV2. GO analysis of all the hits is summarised is Dataset EV3. We find marked enrichment of genes classified in '*Biological processes*' involving DNA ('*DNA recombination*', '*DNA repair*', '*DNA metabolism*') and cell proliferation ('*Mitosis*', '*Cell cycle regulation*'); highlighted within the dotted red box. (B) A plot of the *P* values (Fisher's Exact Test) following GO analysis of the most enriched CYPA-BioID-derived target genes.

activity is required for resection (Fig. 6A). To confirm the role of CYPA specifically in this process, we assessed HU-induced pRPA2 formation in the isogenic CYPA-KO and CYPA-R55A U2OS panel, and similarly observed a markedly reduced signal in both the KO and R55A lines, relative to the scrambled (Scrm)-control (Fig. 6B). Collectively, these findings strongly indicate that loss and/or inhibition of CYPA impairs resection under conditions of replication fork stalling.

## CYPA and RAD51-mediated HRR engagement

The reduction in HRR observed using the DR-GFP system following siCYPA (Fig. 3C) and CYPA PPI inhibition with CsA and NIM811 (Fig. 3D) is consistent with the impaired resection phenotype observed here using pRPA2 (Fig. 6A,B). A resection failure should consequently result in reduced RAD51-mediated strand exchange and D-loop formation, therein underlying the compromised HRR. Consistent with this, we observed a marked impairment in HU-induced RAD51 foci formation in the CYPA-KO and CYPA-R55A cells, in contrast to their isogenic scrambled (Scrm)-control cells (Fig. 6C). Therefore, we find that CYPA loss and inhibition impairs end resection following DNA replication fork stalling, which consequently manifests as reduced RAD51-dependent HRR (Fig. 3C). These represent new phenotypes caused by CYPA loss and inhibition.

## CYPA and the MRN complex

The MRN complex plays an important role in DSB end tethering and in the initiation of resection through interaction with CtIP (Paull, 2018; Syed and Tainer, 2018). NBS1 was found as a candidate 'hit' in the CYPA-BioID, but *not* MRE11 or RAD50 (Dataset EV1; Fig. 5A). We could confirm the co-IP of NBS1 with CYPA, and we found this interaction to be sensitive to treatment with CsA, suggesting the PPI activity of CYPA is required for this interaction (Fig. 7A).

Interestingly, our CYPA IP contained both MRE11 and RAD50, with each respective interaction also being similarly sensitive to CsA (Fig. 7A, dotted red box). BioID is a proximity interacting technique with a purported labelling radius of ~10 nM, and since CYPA at ~18 kDa is dwarfed in size compared to each of the individual MRN components (MRE11: ~70–90 kDa, RAD50: ~150 kDa, NBS1: 95 kDa), we postulated that as only NBS1 was detected in the CYPA-BioID, perhaps it was a direct interactor with CYPA. Furthermore, we reasoned that any putative interacting site must be located away from interacting interfaces between NBS1 and the other MRN components. MRN structural and functional work strongly suggests the N-terminal region of NBS1 fulfills this criterion as it does not bind with MRE11 or RAD50 and protrudes from the core MRN structure, thereby crucially also offering accessibility (Lloyd et al, 2009; Rotheneder et al, 2023; Syed and Tainer, 2018; Williams et al, 2009).

Interestingly, the N-terminal region of NBS1 constitutes an important and indeed seemingly unique constellation of phospho-binding motifs organised as FHA-BRCT$_1$-BRCT$_2$ (Fig. 7B). AlphaFold Protein Structure Database prediction clearly demonstrates the structured nature of this region in human NBS1 (UniProt: O60934) (Fig. 7B,C). It also usefully indicates which proline residues within are likely to be accessible. Verified CYPA target prolines are typically located within disordered/unstructured regions (e.g., as in PRLR and SNCA: Appendix Fig. S2), and sometimes between areas of structure (e.g., as in ITK, CRK and CD147: Appendix Fig. S2). Only a single candidate proline is similarly positioned in NBS1 N-terminus; namely P112, which resides in a very short unstructured linking peptide between FHA and BRCT$_1$ (Fig. 7C). P112 and its neighbouring residues are highly conserved (Appendix Fig. S3). In addition, the E111–P112 containing linker lies immediately downstream of one of the FHA phospho-threonine (pT) binding loops (Fig. 7C Loop #3 composed of -G100-V101-F102-G103-S104-). Whilst alteration of P112 to glycine (P112G) is not predicted to grossly alter the structural order of its immediate environ (Appendix Figs. S4–S6), interestingly, we nonetheless find that altering P112 is predicted not to be tolerated

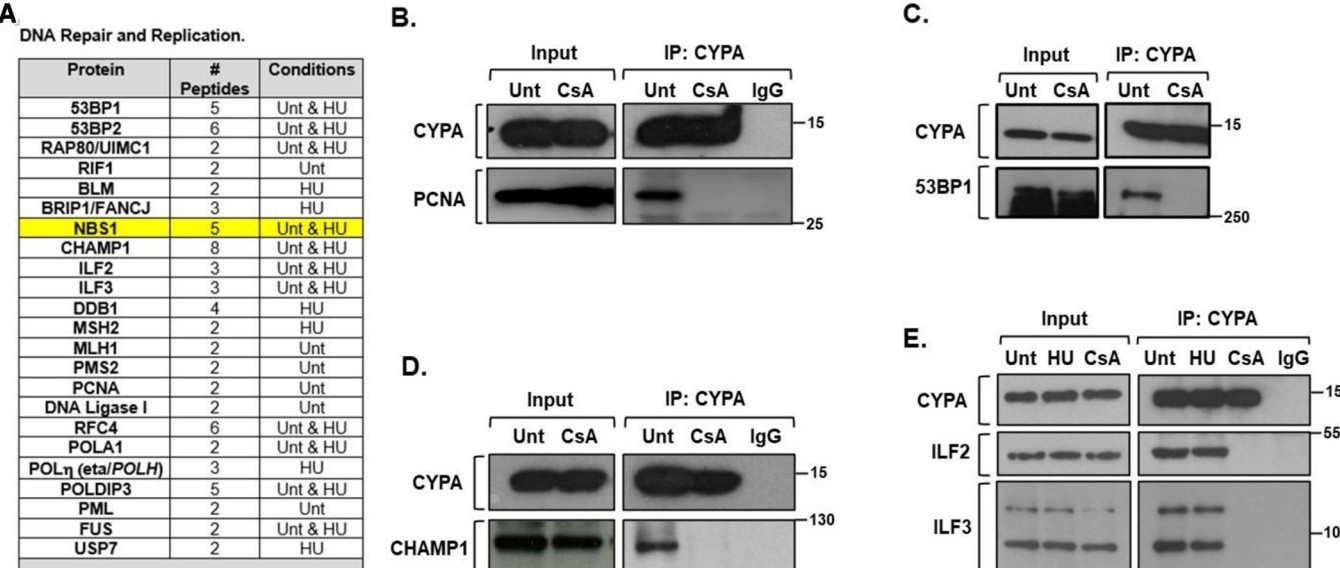

**Figure 5. CYPA interacts with core components of resection mediated homology-directed DNA repair.**

(A) A list of the CYPA-BioID hits that function in *'DNA repair and DNA replication'*. Specific proteins were detected in untreated (Unt) cells, following HU (1 mM, 18 h) or both conditions, as indicated. NBS1, a key focus of this manuscript is highlighted in yellow, was detected in Unt and HU-treated cells. We set out to validate a select set of these interactions by co-immunoprecipitation (co-IP). (B) Endogenous CYPA co-IPs with PCNA. HEK293 cells were either untreated (Unt) or treated with CsA (20 µM, 3 h) prior to IP with anti-CYPA. IgG refers to non-specific immunoglobin control IP. CYPA stably co-IPs PCNA from cycling cells. Interestingly, this interaction is ablated following inhibition of the PPI function of CYPA with CsA, indicating the absolute requirement of this function to sustain this interaction. (C) Endogenous CYPA co-IPs with 53BP1. HEK293 cells were either untreated (Unt) or treated with CsA (20 µM, 3 h) prior to IP with anti-CYPA. Again, we find this interaction is ablated following inhibition of the PPI function of CYPA with CsA, indicating the requirement of this function to sustain this interaction. (D) Endogenous CYPA co-IPs with CHAMP1. HEK293 cells were either untreated (Unt) or treated with CsA (20 µM, 3 h) prior to IP with anti-CYPA. IgG refers to non-specific immunoglobin control IP. Similar to PCNA and 53BP1, this CYPA interaction with CHAMP1 is ablated by CsA, indicating the requirement of the PPI function of CYPA to sustain this interaction. (E) We also find that endogenous CYPA co-IPs with the ILF2-ILF3 complex. HEK293 cells were either untreated (Unt), treated with HU (1 mM, 16 h) or with CsA (20 µM, 3 h) prior to IP with anti-CYPA. IgG refers to non-specific immunoglobin control IP. Co-treatment with HU had no impact on the co-IP. In contrast, CsA impairs CYPA interaction with the ILF2-3 complex, similar to our findings with co-IP of PCNA, 53BP and CHAMP1. These data again indicate a requirement of the PPI function of CYPA to sustain a specific protein-protein interaction. Source data are available online for this figure.

(Appendix Table S3). Indeed except for a single instance (i.e., E111*), variants within the linker peptide have not been reported in cancer, in stark contrast to FHA and $BRCT_1$ (Appendix Table S4).

The AlphaFold-derived structure of human NBS1 suggests that 180° rotation of the peptide bond between E111 and P112 (as occurs in *cis-trans* isomerisation) would impact several important H-bonding interactions between E111 and the extreme N-terminal residues of NBS1, specifically with M1 and K3 (Fig. 7D). Mutating N-terminal residues has been shown to compromise MRN function by disrupting important stacking interactions that consequently alter the structure of the FHA (Williams et al, 2009). Provocatively, modelling binding between CYPA and a peptide containing the linker region between FHA and $BRCT_1$ reveals putative H-bond interactions between E111 and V114 of the linker peptide with key CYPA active site residues (i.e., R55 and Q63) (Fig. 7E).

## CYPA binding to NBS1 and Pro112

To determine if CYPA binds to NBS1 directly, and if so, to identify a specific proline residue that mediates this interaction, we undertook complementary approaches involving ectopic over-expression of full-length FLAG-tagged human NBS1 in HEK293 cells along with bacterial expression of a recombinant HIS-tagged human NBS1 peptide composed of FHA-*linker*-$BRCT_1$ *(aa 1–182)*.

Using HEK293 cells, following expression of full-length FLAG-NBS1 (incl. WT, P112G or P64G) and IP-ing with anti-FLAG, we found endogenous CYPA was co-IP'd with NBS1-WT, in contrast to NBS1-P112G (Fig. 8A, lower panel). We found that NBS1-P64G could also co-IP with endogenous CYPA under these conditions, indicating that ablation of a different proline within the FHA did not compromise the interaction with CYPA (Fig. 8A, lower panel). Our findings suggest that CYPA can interact with NBS1, and that this interaction is disrupted by altering the sole proline within the very short linking region between FHA and $BRCT_1$; namely P112.

To demonstrate direct protein binding, we independently expressed human Strep-tagged CYPA and HIS-tagged human NBS1 FHA-$BRCT_1$ peptide in the bacterial system, incubated them together, and pulled down interacting proteins using Strep beads. Using this approach, we found that CYPA interacts directly with wild-type (WT) human NBS1 FHA-$BRCT_1$ peptide, whilst it fails to interact with the P112G counterpart (Fig. 8B, dotted red box). These data indicate that CYPA can directly interact with FHA-$BRCT_1$ of human NBS1 and that P112 is required to sustain this interaction.

## CYPA influences NBS1 and MDC1 foci formation and NBS1-P112 is required for DNA repair

Our findings strongly suggest a physiologically significant relationship via direct interaction between the PPI CYPA and NBS1 in

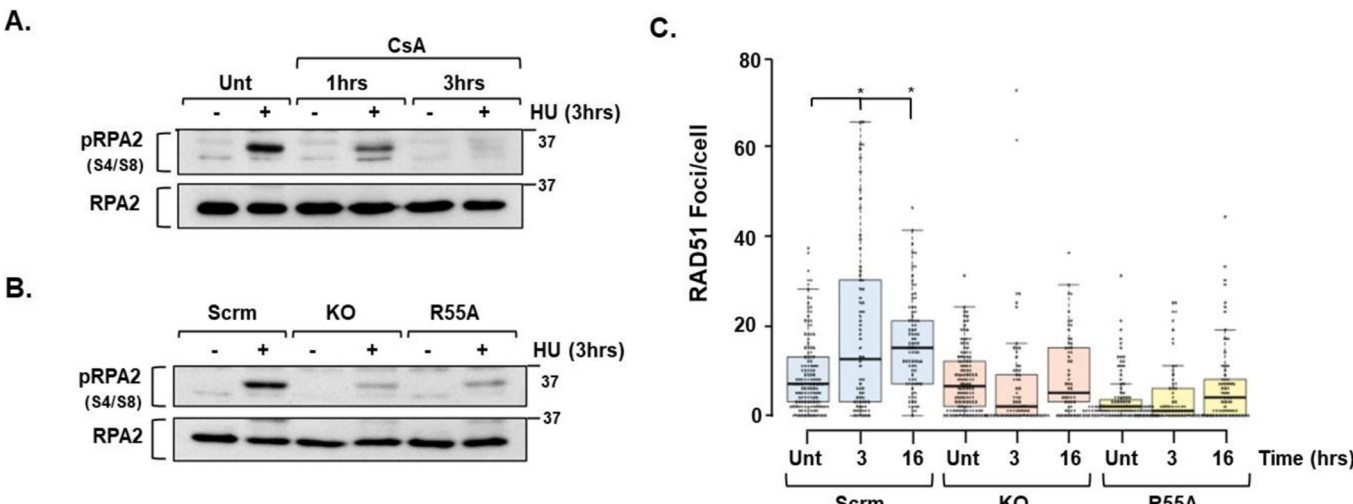

**Figure 6.   CYPA loss and inhibition results in reduced ssDNA and RAD51 foci formation following DNA replication fork stalling.**

(A) U2OS cells were left untreated (Unt) or pre-treated (1 h or 3 h) with CsA (20 μM) before the addition of HU (2 mM, 3 h) in some instances and then harvested. Whole-cell extracts were western blotted for pRPA2 (S4/S8) to indicate ssDNA formation. Only untreated (Unt) cells treated with HU (2 mM, 3 h) produced a robust pRPA2 signal compared to their completely untreated counterparts. Some HU-induced pRPA2 signal was observed at 1 h in CsA, but this was completely lost at 3 h. These data indicate that CsA impairs HU-induced pRPA2 formation. (B) Using the engineered U2OS isogenic panel of lines described in Fig. 1C, including scrambled control (Scrm), *PPIA*/CYPA knockout (KO) and KO reconstituted with a PPI-dead CYPA variant (R55A), we found significantly reduced pRPA2 formation following HU (2 mM, 3 h) in both the KO and R55A lines, in marked contrast to the Scrm control. This indicates that CYPA and/or its PPI activity are required for optimal pRPA2/ssDNA formation following treatment with HU. (C) Indirect immunofluorescence (IF) quantification of RAD51 foci formation in scrambled control (Scrm), *PPIA*/CYPA knockout (KO) and PPI-dead (R55A) isogenic U2OS cell lines following treatment with HU (2 mM for 3 h and 1 mM for 16 h). We observed a significantly increased number of RAD51 foci/cell following HU in the Srm control cells, in stark contrast to the KO and R55A lines. Box boundaries are defined by the 25th and 75th percentiles. The whiskers extend to the minimum and maximum values. Median values indicated by the horizontal lines are *Scrm* [Unt median = 7, 3 h median = 12.5, 16 h median = 15], KO [Unt median = 6.5, 3 h median = 2, 16 h median = 5], R55A [Unt median = 2, 3 h median = 1, 16 h median = 4], *Scrm 3 h $P$ = 0.0001, *16 h $P$ = 2.047 × 10$^{-6}$, Student's $t$ test.). Scrm Unt $n$ = 137, 3 h $n$ = 70, 16 h $n$ = 85, KO Unt $n$ = 122, 3 h $n$ = 73, 16 h $n$ = 61, R55A Unt $n$ = 112, 3 h $n$ = 67, 16 h $n$ = 66. Source data are available online for this figure.

human cells. NBS1 FHA-BRCTs are essential in mediating MRN recruitment and the effective execution of HRR, and other DSB repair pathways (Cerosaletti and Concannon, 2003; Sakamoto et al, 2007; Tauchi et al, 2001; Zhao Renthal and Lee, 2002). Consistent with this, we find both NBS1 foci formation and those of its constitutive interacting partner MDC1 (Chapman and Jackson, 2008; Melander et al, 2008; Spycher et al, 2008), to be significantly impaired/unresponsive in CYPA-KO and CYPA-R55A U2OS lines following DNA replication fork stalling (Fig. 9A,B).

To specifically investigate the impact of NBS1-P112 on DNA repair we expressed FLAG-NBS1 constructs (WT and P112G) in NBS-ILB1 Nijmegen breakage syndrome patient-derived fibroblasts (Fig. 9C), and examined these lines for HU-induced NBS1 and RAD51 foci formation (Fig. 9D,E). In contrast to the expression of NBS1-WT, we observed impaired HU-induced NBS1 (Fig. 9D) and RAD51 (Fig. 9E) foci formation following NBS1-P112G expression, indicating that mutation of P112, the residue required to sustain interaction with CYPA, results in impaired execution of DNA repair following replication fork stalling.

Collectively, these findings could explain the specific defective DNA repair profile (Fig. 3), the impaired resection (Fig. 6A,B) and RAD51 foci formation (Fig. 6C) we observed following CYPA loss and inhibition. The aberrantly altered foci formation are also consistent with a functionally relevant interaction between CYPA and NBS1 than can influence MRN activity following replication fork stalling (Figs. 7 and 8). Expression of the MRN complex and the sub-cellular distribution of NBS1 was unaffected by CYPA-KO

or CYPA-R55A expression compared to Scrm control (Appendix Fig. S7A,B). Unfortunately, CYPA does not form foci, precluding further IF-based co-localisation studies (Appendix Fig. S7C,D).

## Impaired CYPA function reveals novel genetic dependencies/vulnerabilities

CtIP is a master upstream regulator of DNA repair pathway choice by functioning as an interacting platform for partners such as MRN (Andres and Williams, 2017; Cejka and Symington, 2021; Makharashvili and Paull, 2015; Mozaffari Pagliarulo and Sartori, 2021; Reginato and Cejka, 2020). We reasoned that if CYPA loss and/or inhibition compromises MRN functionality, the viability of CYPA-KO and CYPA-R55A cells should be over-dependent on the principal regulator of DNA repair pathway choice; namely, CtIP. We observed marked loss of clonogenic survival in CYPA-KO and CYPA-R55A cells following siCtIP in the absence of additional exogenously supplied DNA damage, in contrast to their isogenic wild-type scrambled (Scrm)-controls (Fig. 10A). This demonstrates that CYPA-defective cells are strongly dependent upon CtIP for viability.

The DNA repair defects revealed by the DR-GFP-reporter analyses indicate a relatively milder impact of CYPA loss and inhibition on SSA, compared to HRR (Fig. 3B,C). This suggests that CYPA-compromised cells may also be specifically overly dependent upon residual functional SSA for survival. This dependency would also be consistent with compromised MRN complex function whose exonuclease activity is required to initiate use of this

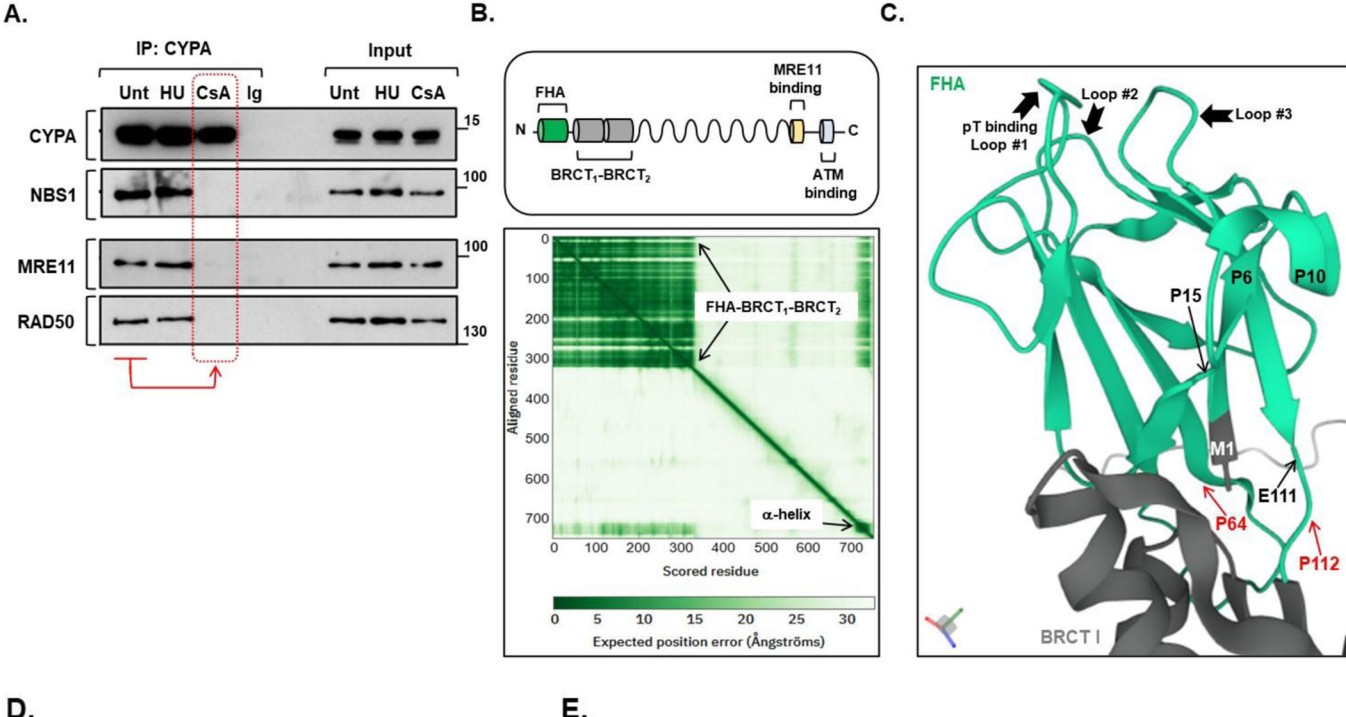

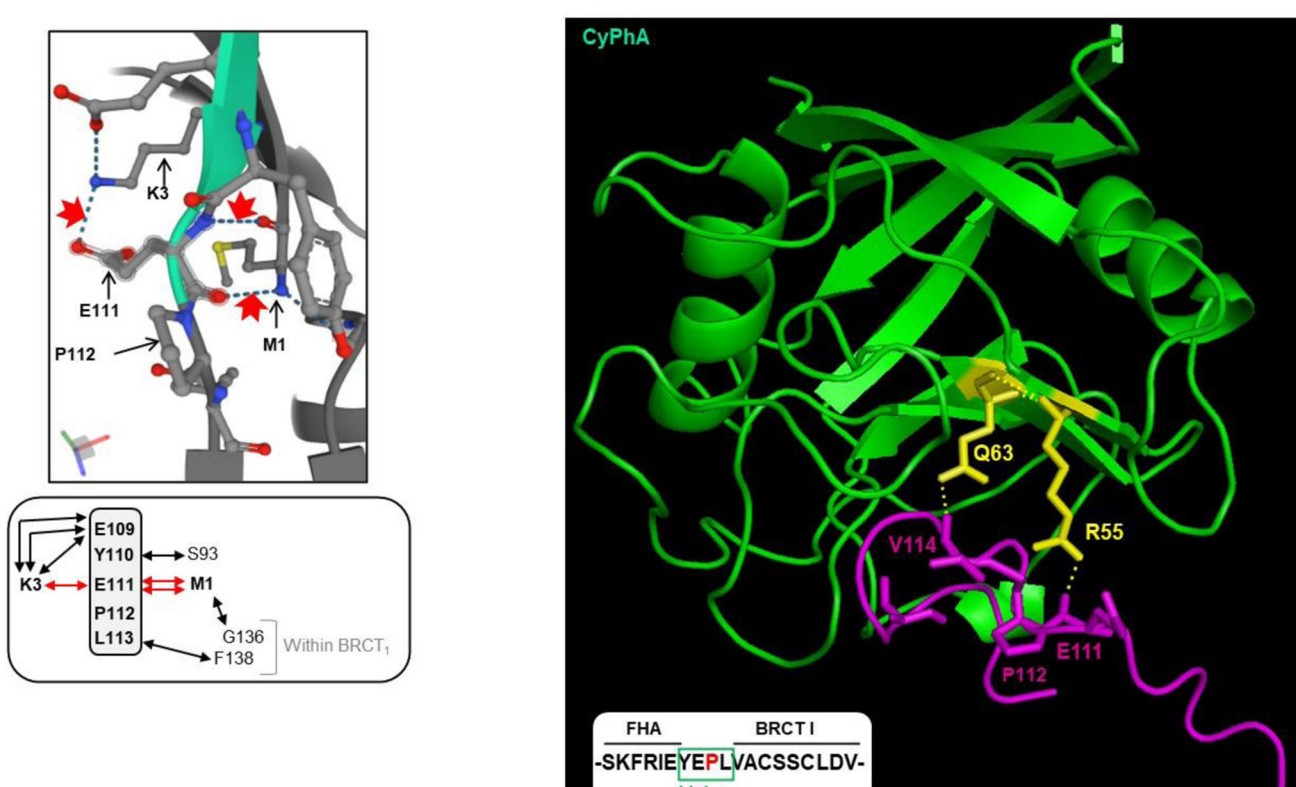

pathway. Indeed, we found that siRAD52 to be strongly lethal in CYPA-KO and CYPA-R55A cells (Fig. 10B). In addition, and in agreement with the necessity for CYPA PPI function, we found that siRAD52 results in hypersensitivity to killing by CsA in U2OS (Fig. 10C).

CYPA loss and/or inhibition causes reduced DNA replication, reduced fork speed, and increased levels of HU-induced fork stalling (Fig. 2A–C). The basis of these phenotypes is likely multifactorial. For example, several DNA replication and cell cycle factors were found in the CYPA-BioID (Fig. 5A; Datasets EV1–3).

**Figure 7. The structured N-terminus NBS1 is a likely CYPA interacting region.**

(A) Endogenous CYPA co-IPs with the MRN complex in a manner dependent upon CYPA's PPI activity. CYPA was IP'd from HEK293 cells that were untreated (Unt) or treated with HU (1 mM, 16 h) or CsA (20 μM, 3 h). A non-specific immunoglobin (IgG) was used as a control IP. CYPA was found to efficiently co-IP NBS1, MRE11 and RAD50 from Unt and HU-treated cell extracts. This contrasted with CsA-treated cell extracts, where interaction with all three MRN complex components was inhibited (indicated by the red dotted box). This shows that the PPI activity of CYPA is required to sustain interaction with the MRN complex. Treatment with HU did not disrupt the observed interaction. (B) The upper panel depicts the primary structure of the NBS1 component of the MRN complex, showing the relative positioning of its principal functional domains. In its N-terminus, the juxtaposition of the FHA with the two BRCTs is unique to NBS1. The lower panel shows the predicted aligned error plot generated by AlphaFold for NBS1 UniProt: O60934, clearly highlighting the main area of structure as the FHA-BRCT$_1$-BRCT$_2$ (aa 1–325). (C) This panel shows a close-up image of the FHA domain (in green) together with a small portion of BRCT$_1$, as generated by AlphaFold. The loops implicated in phospho-threonine binding are indicated (Loop #1–3), along with the key residues discussed in the associated text (i.e., M1, P64, E111 and P112). NBS1 FHA contains P6, P10, P15 and P64; all of which are indicated. (D) The upper panel shows a close-up image of the AlphaFold generated structure of the extreme N-terminus of NBS1 with various amino acid sidechains revealed, demonstrating how it is connected to E111 in the linker region through H-bonding. The H-bonds discussed in the text are highlighted by the red arrows (i.e., 2× for M1-E111 and 1× for K3-E111). The lower panel is a summary of all local H-bond interactions for the linker region (indicated by the double arrows). The H-bond interactions with E111 are highlighted in red. (E) This shows an interaction model generated using the CoLabFold v1.5.2-patch (AlphaFold2 using MMseqs2) between CYPA (green) and an NBS1-derived peptide (magenta), composed of -SKFREYEPLVACSSCLDV-, incorporating the linker region (boxed) and representing a tract of 100% conservation at both sides of the linker (Appendix Fig. S3). H-bond interactions between important residues in the CYPA active site implicated in prolyl isomerisation catalysis (i.e., R55 and Q63; sidechains highlighted in yellow) and linker residues in the NBS1 peptide, including E111, were predicted and are indicated by the dotted lines. Source data are available online for this figure.

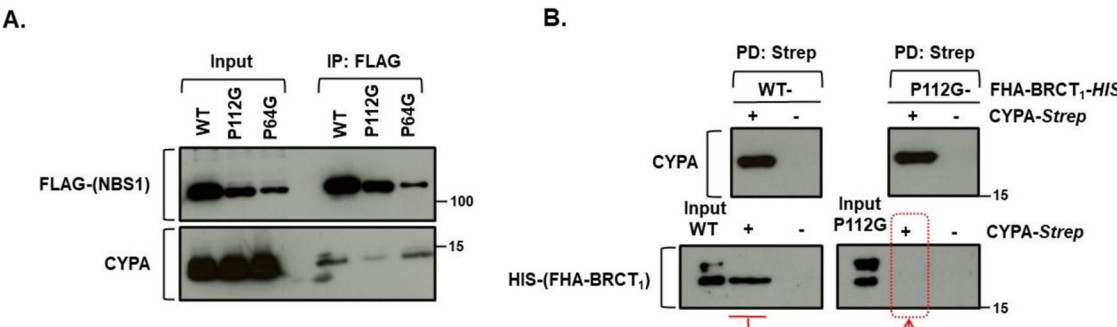

**Figure 8. CYPA's direct interaction with NBS1 N-terminus is mediated by P112.**

(A) The upper panel shows the relative expression of various FLAG-tagged full-length human NBS1 constructs (WT, P112G, P64G) following transient ectopic overexpression in HEK293 cells (Input) and after IP using anti-FLAG beads. P64G and P112G do express, although to a lesser degree than that of WT NBS1. The lower panel shows CYPA western blot in the input and following FLAG-IP. CYPA co-IP following expression of NBS1-P112G is markedly impaired compared to that of wild-type (WT) NBS1 and NBS1-P64G, suggesting that P112 is required to mediate interaction between full-length NBS1 and CYPA. (B) Recombinant wild-type (WT) HIS-tagged FHA-BRCT$_1$ and P112G HIS-tagged FHA-BRCT$_1$ were mixed with CYPA-Strep or empty beads, before pulldown with MagStrep "type 3" XT Beads (*IBA Lifesciences GmbH*), to pulldown Strep-tagged CYPA. The upper western blot panels show the recovery of CYPA-Strep by the ManStrep XT beads when it was included in the incubations. The lower western blot panels show that whilst WT HIS-tagged FHA-BRCT$_1$ could be recovered following Strep capture, indicating a direct interaction with CYPA, no P112G HIS-tagged FHA-BRCT$_1$ was obtained when co-incubated with the CYPA-Strep (red arrow and dotted box). This indicates that ablation of P112 impairs the direct interaction with CYPA. Source data are available online for this figure.

Fascinatingly, these DNA replication phenotypes were associated with relative resistance to HU-induced killing in the CYPA-KO and CYPA-R55A cells (Fig. 11A). Interestingly, impaired ATR/NBS1-dependent pRPA2 has been shown to result in resistance to killing by HU (Manthey et al, 2007). We postulated that this could be reflective of some form of stalled replication fork protection process being engaged when CYPA function is compromised. If this were the case, removing these fork protection factors could result in loss of viability in this context.

The RAD51 paralog family have integrating roles in fork protection, fork recovery and Holliday Junction formation (Berti et al, 2020; Bhattacharya et al, 2022; Bonilla et al, 2020). The paralogues function as two distinct complexes, BCDX2 (i.e., RAD51B-RAD51C-RAD51D-XRCC2) and CX3 (i.e., RAD51C-XRCC3). We found that Xrcc3-defective irs1SF CHO cells to be profoundly sensitive to killing by CsA, compared to their AA8 parental line (Fig. 11B). Similarly, siXRCC3 resulted in reduced clonogenic survival of both the CYPA-KO and CYPA-R55A

U2OS cell lines, indicating that the CX3 complex is required for the viability of these human cells (Fig. 11C). RAD51C is the only member common to both paralogue complexes; BCDX2 and CX3. We found that siRAD51C resulted in increased loss of clonogenic survival of the CYPA-KO and CYPA-R55A lines, relative to their isogenic scrambled (Scrm)-control (Fig. 11D). The effects of targeting the BCDX2 and CX3 complexes were in stark contrast to that of silencing the fork remodelling translocases ZRANB3 and SMARCAL1, whereby silencing of each did not selectively impact the viability of CYPA-KO or CYPA-R55A lines (Appendix Fig. S8A,B). Indeed this was also true for siSWSAP1 (a different RAD51 paralog), siWRN, siREF1/APE1, siAPE2 and siHMCES (Appendix Fig. S8C–G). Finally, BRCA2, in addition to its role in HRR also plays an important function in fork protection (Feng and Jasin, 2017; Kolinjivadi et al, 2017; Prakash et al, 2015). As anticipated from our siXRCC3 and siRAD51C-derived observations and consistent with a dependence upon fork protection under conditions of impaired CYPA, we found that conditional

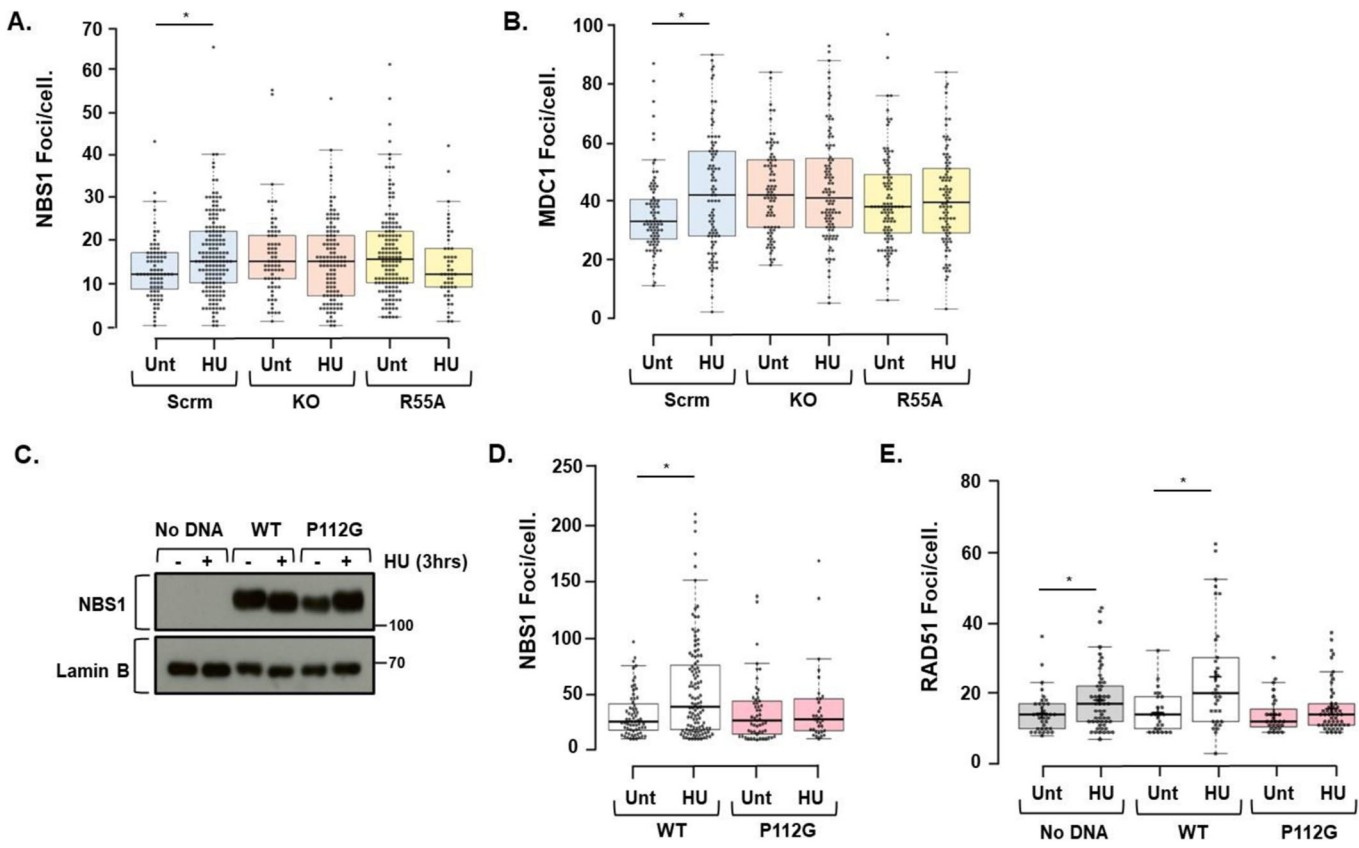

**Figure 9. CYPA loss, inhibition and P112G results in impaired NBS1 and MDC1 foci formation.**

(A) Indirect IF of NBS1 foci formation in the isogenic U2OS panel (Scrm; scrambled control, KO; *PPIA*/CYPA knockout, R55A; KO reconstituted with CYPA p.R55A) following treatment with HU (2 mM, 3 h). A significant increase in the formation of NBS1 foci/cell following HU was only found in the scrambled control cells (*P = 0.0022, Student's *t* test, Unt n = 80, HU n = 162). Interestingly, we did not observe an increase in HU-induced NBS1 foci/cell in either the KO or R55A. Moreover, in these cells there appeared a spontaneously elevated yet unresponsive amount of NBS1 foci/cells when compared to Scrm. Box boundaries are defined by the 25th and 75th percentiles. The whiskers extend to the minimum and maximum values. The median values for each treatment dataset are indicated on the bar chart by the horizontal black lines (Scrm Unt median = 12 and HU median = 15. KO Unt median = 15 and HU median = 15. R55A Unt median = 15.5 HU median = 12). (B) Indirect IF of MDC1 foci formation in the U2OS-engineered cell line panel either untreated (Unt) or following HU (2 mM, 3 h). Here again, a significant increase in the formation of HU-induced MDC1 foci/cell was only found in the scrambled control cells (*P = 0.0069, Student's *t* test, Unt n = 83, HU n = 88). We did not observe a response to HU treatment in the KO or R55A cell lines. Interestingly, as for NBS1 foci, MDC1 foci/cell were found spontaneously elevated yet unresponsive to HU in the KO and R55A cell lines. Box boundaries are defined by the 25th and 75th percentiles. The whiskers extend to the minimum and maximum values. The median values for each treatment dataset are indicated on the bar chart by the horizontal black lines (Scrm Unt median = 33 and HU median = 42. KO Unt median = 42 and HU median = 41. R55A Unt median = 38 HU median = 39.5). (C) Expression of NBS1-WT and NBS1-P112G in NBS-ILB1, a Nijmegen Breakage Syndrome (NBS) patient-derived fibroblast. This line is homozygous for the *NBN* founder mutation (657del5), p.Lys219Asn*fs*16 (c.657_661delACAAA) and does not expression full-length NBS1 (No DNA, mock transfection). (D) Indirect IF of NBS1 foci formation in NBSI-LB1 cells transfected with NBS1-WT or NBS-P112G, either untreated (Unt) or following treatment with HU (2 mM, 3 h). A significant increase in HU-induced NBS1 foci formation was observed following expression of NBS1-WT in contrast to expression of NBS1-P112G (*P = 0.0002, Student's *t* test, WT Unt n = 64, WT HU n = 111, P112G Unt n = 53, P112G HU n = 33). Box boundaries are defined by the 25th and 75th percentiles. The whiskers extend to the minimum and maximum values. Median values indicated by the horizontal lines (WT Unt median = 26, WT HU median = 39, P112G Unt median = 27, P112G HU median = 28). (E) Indirect IF of RAD51 foci formation in NBS-ILB1 cells mock-transfected (No DNA), transfected with NBS1-WT or NBS-P112G, either untreated (Unt) or following treatment with HU (2 mM, 3 h). Significant increases in HU-induced RAD1 foci formation were observed following mock transfection (No DNA) and expression of NBS1-WT, in contrast to expression of NBS1-P112G. Box boundaries are defined by the 25th and 75th percentiles. The whiskers extend to the minimum and maximum values (*No DNA P = 0.0220, *WT P = 0.0010, Student's *t* test, No DNA Unt n = 36, No DNA HU n = 53, WT Unt n = 25, WT HU n = 34, P112G Unt n = 28, P112G HU n = 52). The median values for each treatment are No DNA Unt median = 14, No DNA HU median = 17, WT Unt median = 14, WT HU median = 20, P112G Unt median = 12, P112G HU median = 14. The *NBN* founder mutation in NBS-ILB1 is hypomorphic, which likely explains the modest increase in RAD51 foci formation in the No DNA mock-transfected cells. The mean number of RAD51 foci/cell is nonetheless reduced in these cells compared to those expressing NBS1-WT (No DNA Unt mean = 14.5, No DNA HU mean = 18.11 compared to WT Unt mean = 14.6, WT HU mean = 24.79), and is more like that observed following NBS1-P112G expression (P112G Unt mean = 13.96, P112G HU mean = 15.85). Source data are available online for this figure.

shBRCA2 also resulted in hypersensitivity to killing by CsA (Fig. 11E).

Therefore, we propose that CYPA loss and/or inhibition impairs optimal engagement of key DSB repair pathways by principally compromising end resection and this is associated with an additional set of genetic dependencies upon key components of these interconnected pathways including CtIP (SSA, HRR), RAD52 (SSA) and LIG4 (NHEJ). In addition, we find that CYPA loss and/or inhibition results in loss of viability when important proteins active in DNA replication fork protection are downregulated,

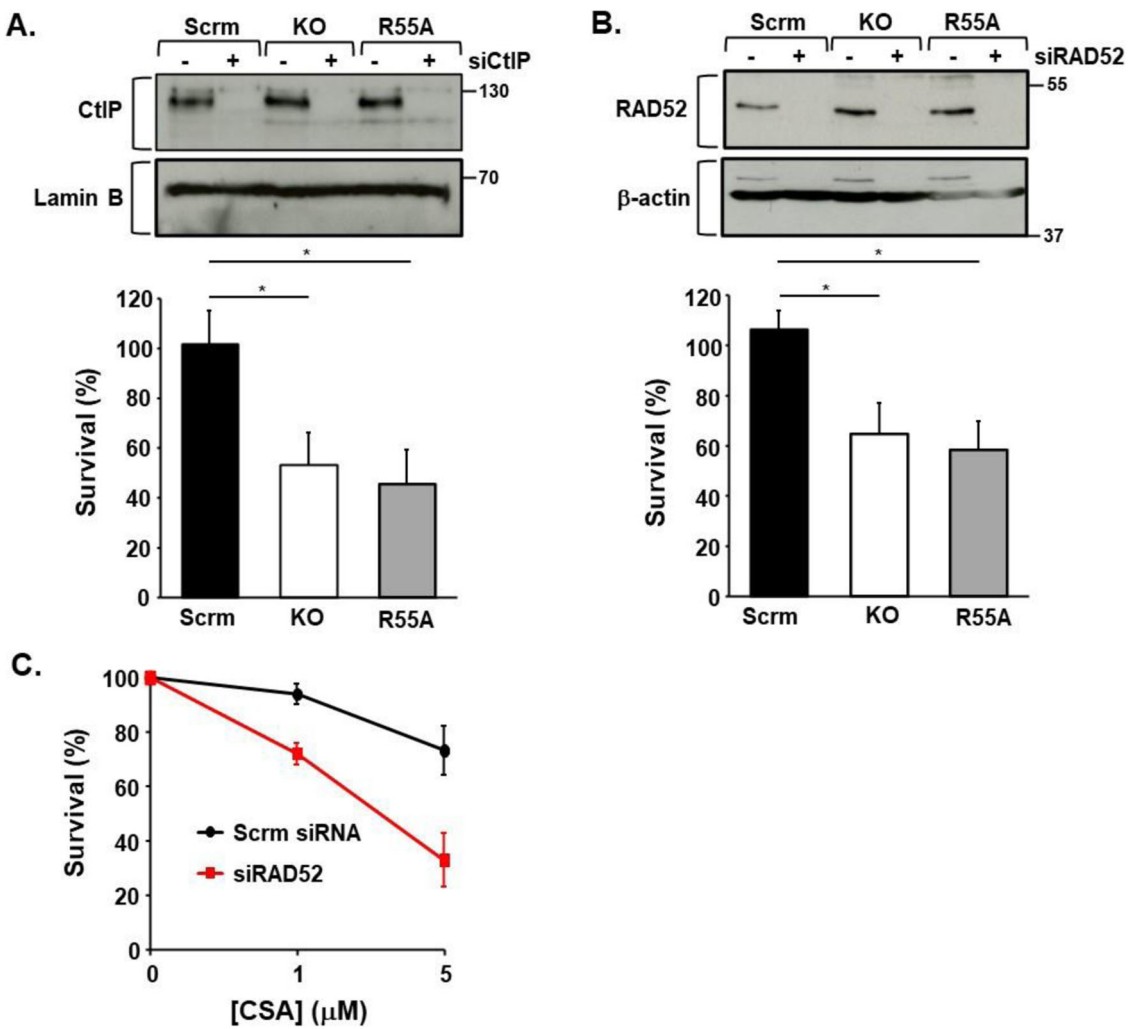

**Figure 10.   Cells with absent and inhibited CYPA are dependent upon CtIP and RAD52 for viability.**

(A) The upper panel shows western blot analysis of CtIP expression following siRNA of CtIP (siCtIP) in the U2OS panel (Scrm; scrambled control, KO; PPIA/CYPA knockout, R55A; KO reconstituted with CYPA p.R55A). The bar chart summarises the survival of these cells (7 days) following siCtIP, as determined by crystal violet staining and extraction. In the absence of exogenously supplied DNA damage, the survival of both the KO and R55A lines is significantly compromised following transient siCtIP (error bars represent the mean ± s.d. *KO P = 0.0380, *R55A P = 0.0328, Student's t test, n = 3 independent determinations). These data show that CtIP is required for viability in the absence of CYPA and following inhibition of CYPA PPI function. (B) The upper panel shows western blot analysis of RAD52 expression following siRNA of RAD52 (siRAD52) in the same U2OS-engineered cell line panel. The bar chart summarises the survival of these cells (7 days) following siRAD52, as determined by crystal violet staining and extraction. In the absence of exogenously supplied DNA damage, the survival of both the KO and R55A lines is significantly reduced following transient siRAD52 (error bars represent the mean ± s.d. *KO P = 0.0349, *R55A P = 0.0431, Student's t test, n = 3 independent determinations). These data indicate that RAD52 is also required for viability in the absence of CYPA and following inhibition of CYPA PPI function. (C) Consistent with our findings in the U2OS R55A cells following siRAD52, we find that wild-type U2OS cells are more sensitive to killing by CsA following siRAD52 compared to those treated with a non-targeting scrambled siRNA (error bars represent the mean ± s.d. of n = 3 independent experiments). Source data are available online for this figure.

including the RAD51 paralogues XRCC3 and RAD51C, as well as BRCA2. Collectively, these represent completely novel insights into CYPA biology.

## Potential rational applications of CYPA inhibition in select cancers

The question then arises as to whether any of these novel insights could be leveraged towards a translational application? After all, effective pharmacological CYPA inhibitors already exist. Interestingly, DepMap *Cancer Gene Dependency* data show that reduced

viability (fitness) following *PPIA*/CYPA-KO is more commonly seen in breast cancer cell lines with reduced *BRCA2* copy number (Appendix Table S5).

Amplification of the MYC transcription factor family member *MYCN* drives a range of solid tumours including medulloblastoma, rhabdomyosarcoma, osteosarcoma, Wilms tumour, small cell lung cancer and retinoblastoma, with MYCN overexpression associated with poorer prognoses (Rickman Schulte and Eilers, 2018). Approximately 25% of paediatric neuroblastoma (NBm) is associated with robust *MYCN* amplification (often ≥100× copies), exhibiting elevated replication stress (RS)-mediated genomic

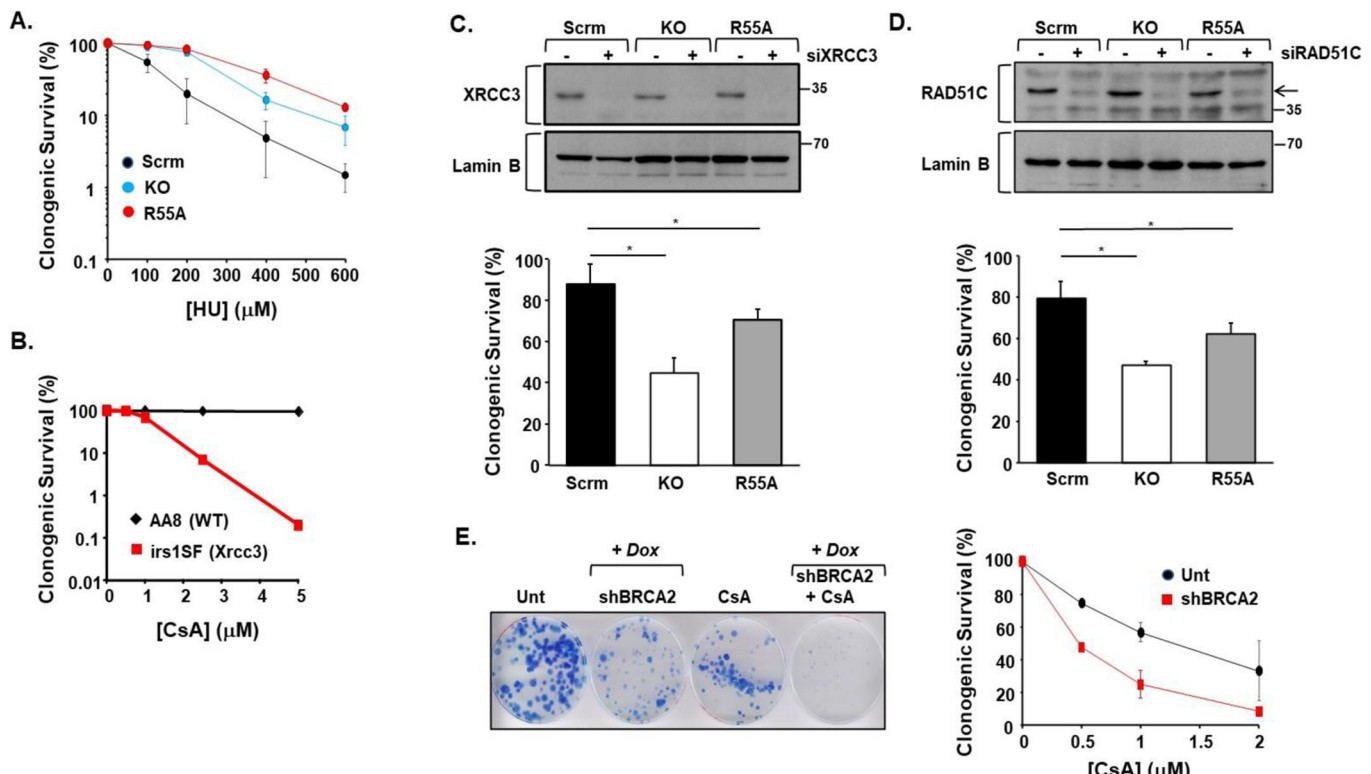

**Figure 11. Cells with absent and inhibited CYPA are dependent upon DNA replication fork protection pathways for viability.**

(A) Clonogenic survival of U2OS engineered panel (Scrm; scrambled control, KO; PPIA/CYPA knockout, R55A; KO reconstituted with CYPA p.R55A) following treatment with increasing doses of HU (error bars represent the mean ± s.d of n = 3 independent determinations). Interestingly, both KO and R55A cells are relatively more resistant to killing compared to Scrm control cells. (B) Clonogenic survival of CHO cells following treatment with increasing doses of CsA. The Xrcc3-defective CHO line irs1SF is markedly more sensitive compared to its parental control line AA8 (WT: wild-type). These data indicate that Xrcc3 is required for survival following CYPA PPI inhibition using CsA. (C) The upper panel shows western blot analysis of the RAD51 paralogue XRCC3 expression following siRNA of XRCC3 (siXRCC3) in the U2OS-engineered cell line panel. The bar chart summarises the clonogenic survival of these cell following siXRCC3. In the absence of exogenously supplied DNA damage, the clonogenic survival of both the KO and R55A lines is significantly reduced following transient siRAD52 (error bars represent the mean ± s.d. *KO P = 0.0009 *R55A P = 0.0454, Student's t test, n = 3 independent determinations). These data indicate that XRCC3 is required for normal clonogenic survival in the absence of CYPA and following inhibition of CYPA PPI function. (D) The upper panel shows western blot analysis of the RAD51 paralogue RAD51C expression following siRNA of RAD51C (siRAD51C) in the U2OS-engineered cell line panel. The bar chart summarises the clonogenic survival of these cell following siRAD51C. In the absence of exogenously supplied DNA damage, as for siXRCC3, the clonogenic survival of both the KO and R55A lines is significantly reduced following transient siRAD51C (error bars represent the mean ± s.d. *KO P = 0.0243, *R55A P = 0.0193, Student's t test, n = 3 independent determinations). These data indicate that RAD51C is required for effective clonogenic survival in the absence of CYPA and following inhibition of CYPA PPI function. (E) The panel depicts the clonogenic survival of non-small cell lung carcinoma cell line H1299 following conditional (Dox: doxycycline)-induced shRNA of BRCA2 (shBRCA2), in the presence and absence of CsA. Unt; untreated and uninduced (non-Dox treated) cells. shBRCA2 was induced for 72 h before addition of CsA (5 μM). Clones were stained (Giemsa) after 2 weeks. shBRCA2 alone results in some loss of clonogenic survival compared to Unt, as expected, as does treatment with CsA (5 μM) alone. Interestingly, CsA treatment of shBRCA2 cells results in a marked reduction in clonogenic survival compared to Unt, the shBRCA2 alone cells and the CsA-treated uninduced cells, indicating that BRCA2 is required for survival following CYPA PPI inhibition. The plot shows the clonogenic survival of the Unt and shBRCA2 cells following treatment with a range of CsA concentrations (error bars represent the mean ± s.d., n = 3 independent determinations). Source data are available online for this figure.

instability, and characterised by a more aggressive and poorly responding form of the disease (Carén et al, 2010; Maris 2010; Pugh et al, 2013). In fact, a *MYCN*-driven NBm transcriptional signature has been defined with altered expression of several genome stability pathway components. This has been invoked for the rational use of PARP inhibition in this context, even progressing to clinical trials (e.g., https://clinicaltrials.gov/ NCT03233204, NCT04901702, NCT04544995, NCT03155620) (Hallett et al, 2016). In addition, the elevated RS-associated with *MYCN* amplification has been demonstrated as a vulnerability following CDK2i, CHEK1i, PARPi, ATRi, DNA-PKi and even for MRE11i in various models of NBm (Cole et al, 2011; Colicchia et al, 2017; Dolman et al, 2015; King et al, 2020; King et al, 2021; Molenaar et al, 2009; Petroni et al, 2018). The latter is particularly relevant in the context of our novel

findings regarding CYPA and MRN-dependent DNA repair described here. Fascinatingly, we find that *MYCN*-amplified NBm cell lines are hypersensitive to killing by CsA, compared to non-*MYCN*-driven NBm lines (Fig. 12A–C). Consistent with our findings, DepMap *Cancer Gene Dependency* data show that *PPIA/CYPA*-KO causes frequent loss of fitness/viability in a large panel of NBm cell lines (Dataset EV4).

Multiple myeloma (MM) is a genetically heterogeneous plasma cell malignancy that primarily affects the elderly (>70 years) (Bolli et al, 2014; Corre Munshi and Avet-Loiseau, 2015; Kumar et al, 2017; Palumbo and Anderson, 2011). MM is a chronic disease often associated with severe comorbidities including lytic bone lesions, increased risk of multiple fractures, anaemia, and compromised immunity (Diaz-delCastillo et al, 2021; Walker et al, 2014). Whilst

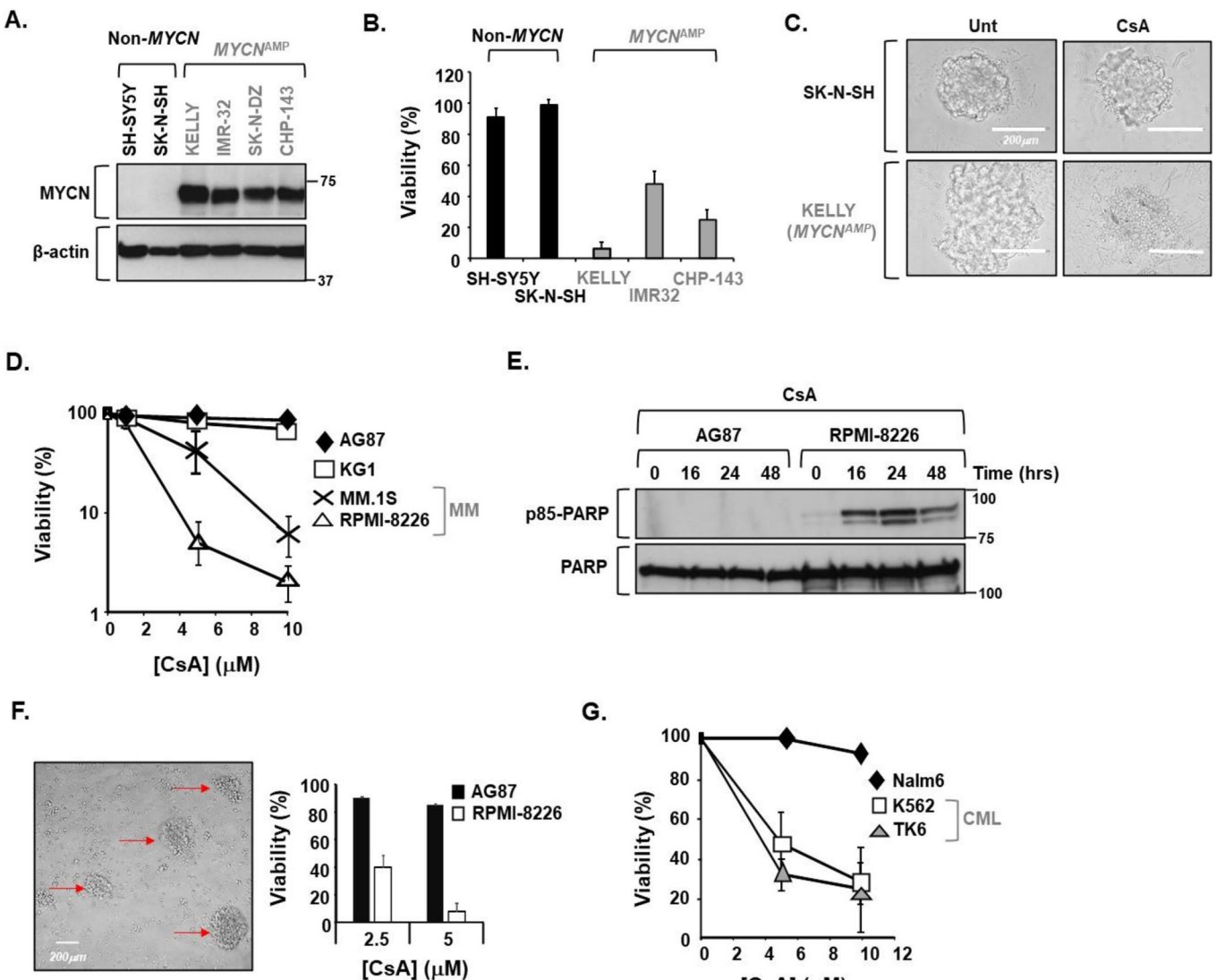

**Figure 12. Select cancer cell lines exhibit elevated sensitivity to killing by the CYPA inhibitor CsA.**

(A) Western blotting of a panel of paediatric neuroblastoma (NBm) cell lines showing MYCN expression in non-*MYCN*-driven NBm (Non-*MYCN*) and from lines with known amplification involving *MYCN* (*MYCN*AMP) (error bars represent the mean + s.d. where $n = 3$ independent experiments). (B) CellTitre Blue viability assay analysis of different NBm cell lines with differing *MYCN* expression following treatment with CsA (10 µM, 72 h), showing elevated sensitivity towards CsA segregating with *MYCN* amplification in this panel of cell lines (error bars represent the mean + s.d. where $n = 3$ independent experiments). (C) Tumour spheroid formation in NBm cell lines untreated (Unt) or following treatment with CsA (10 µM, 72 h) was shown to be completed ablated due to elevated toxicity in the *MCYN*AMP NBm cell line KELLY, in marked contrast to that of the non-*MCYN* driven NBm cell lines SK-N-SH. Scale bar: 200 µm. (D) CellTitre Blue viability assay analysis of Multiple Myeloma (MM) patient-derived cell lines (MM1.S and RPMI-8226) compared to a normal wild-type B-cell lymphoblastoid line (AG87) and KG1 cells, which is a macrophage bone marrow aspirate cell line from an acute myeloid leukaemia patient (error bars represent the mean + s.d. where $n = 3$ independent experiments). Both MM patient-derived lines show elevated sensitivity to killing by CsA compared to AG87 and KG1. (E) Western blot analysis of a CsA (5 µM) time-course using an antibody specific to the p85 cleaved version of PARP, which is an indicator of apoptosis induction, in extracts from a normal wild-type B-cell lymphoblastoid line (AG87) and the MM patient line RPMI-8226. The p85-PARP signal is only detectable in the MM cell line, consistent with increased sensitivity to killing by CsA under these conditions. (F) The image shows 3D-colonies of MM patient line RPMI-8226 in fibrinogen matrix used to mimic the bone marrow niche (red arrows; after 72 h). The associated bar chart shows the survival of these MM colonies and those of a wild-type normal B-lymphoblastoid cell line AG87, following treatment with CsA (72 h) (error bars represent the mean + s.d. where $n = 3$ independent experiments). The MM patient line shows enhanced sensitivity to killing by CsA compared to a wild-type B-lymphoblast under these conditions. Scale bar: 200 µm. (G) CellTitre Blue viability assay analysis of two BCR-ABL-expressing Chronic Myelogenous Leukaemia (CML) patient-derived cell lines (K562 and TK6) and an Acute Lymphoblastic Leukaemia (ALL) patient line (Nalm6), following treatment with increasing concentrations of CsA (at 72 h) (error bars represent the mean + s.d. where $n = 3$ independent experiments). Both BCR-ABL-expressing CML lines show elevated sensitivity to killing under these conditions, compared to the non-BCR-ABL ALL cell line. Source data are available online for this figure.

current rigorous multimodal induction, consolidation and maintenance treatment regimes are effective, MM remains incurable and novel effective interventions are necessitated (Moreau et al, 2021). MM is characterised by an aberrantly elevated HRR landscape, which is thought to drive disease progression and the emergence of therapy resistance (Shammas et al, 2009). Proteosome inhibition is an effective tool against MM and has been shown to reduce this elevated HRR activity, suggesting combination therapy with PARPi particularly in recurrent refractory MM (e.g., NCT01495351 and NCT01326702) (Neri et al, 2011). Similar to *MYCN*-amplified NBm, we find MM patient-derived cell lines to be selectively sensitive to killing by CsA (Fig. 12D–F). Fascinatingly, as part of a multicentre single-arm clinical trial investigating MM resistance (NCT040065789), Cohen et al recently reported that either *PPIA*/CYPA knockout or inhibition using CsA sensitised resistant MM tumour cells to proteosome inhibition, further underscoring the potential clinical impact of CYPA inhibition (Cohen et al, 2021). Indeed, the potential efficacy of CsA against drug-resistant MM had been noted previously (Pilarski et al, 1998; Sonneveld et al, 1992). Consistent with this and our findings, DepMap *Cancer Gene Dependency* data show that *PPIA*/CYPA-KO causes frequent loss of fitness/viability in a panel of MM cell lines (Dataset EV4).

BCR-ABL fusion tyrosine kinase (FTK) has long been linked to HRR activation and the emergence of therapy resistance in Chronic Myelogenous Leukaemia (CML); a scenario similar to MM (Salles et al, 2011; Slupianek et al, 2005; Slupianek et al, 2002; Slupianek et al, 2006; Slupianek et al, 2001). Resistance to effective and commonly employed FTK inhibitors (e.g. Imatinib/Gleevec, Dasatinib/Sprycel) remains a persistently troubling issue and alternate approaches such as targeting RAD51 via novel small molecule inhibitors have been proposed (Bixby and Talpaz, 2011; Zhu et al, 2013). Similarly, targeting RAD52 has been demonstrated as a logical personalised synthetic lethal strategy in a range of FTK driven leukaemia types with associated impaired BRCA1/2 (Cramer-Morales et al, 2013). We found increased sensitivity to killing by CsA in two BCR-ABL-positive CML lines (Fig. 12G). Interestingly, CsA has previously also been shown to potentiate sensitivity to Imatinib in CML (Frydrych Mlejnek and Dolezel, 2009). Consistent with our findings, DepMap *Cancer Gene Dependency* data shows that *PPIA*/CYPA-KO causes frequent loss of fitness/viability in a panel of CML cell lines (Dataset EV4).

It is likely that the novel impacts on DNA repair and the various genetic dependencies we have identified and outlined here concerning CYPA loss and inhibition contribute to the sensitivity to CsA we and others have demonstrated in these distinct cancer types, all of which are commonly linked by a characteristic genomic profile involving elevated RS and a high dependency upon resection-driven HRR. These findings also demonstrate the exciting and largely unexplored potential clinical efficacy of the targeted application of cyclophilin inhibition against select cancers.

## Discussion

Here, we present a comprehensive proximity interactome of the peptidyl-prolyl *cis-trans* isomerase (PPI) Cyclophilin A (CYPA) and outline how we leveraged this to provide new insight into how loss or inhibition of CYPA impairs HRR following DNA replication fork stalling by suppressing end resection. Amongst interactions

with several proteins that function in end resection, we show that CYPA directly interacts with NBS1. We identify an additional set of vulnerabilities/dependencies associated with CYPA loss and inhibition, before showing that pharmacological inhibition of CYPA can selectively kill a diverse set of cancer cell lines with a shared dysregulation of and addiction to HRR.

The genetic vulnerabilities we identified (i.e., LIG4, KU80, CtIP, RAD52, XRCC3, RAD51C, BRCA2), indicate a dependence on multiple distinct, although interconnected, DNA repair pathways following CYPA loss and/or inhibition. Of note, these vulnerabilities were found in the absence of exogenously supplied genotoxins, suggesting a homoeostatic role for CYPA in genome stability in response to endogenously generated DNA lesions. These dependencies are likely reflected in the range of chromosomal abnormalities we observed following treatment with CsA (Fig. 1A,B), where fusion-type events predominate, perhaps suggestive of elevated levels of rearrangements and of 'toxic endjoining' (Britton et al, 2020). The presence of SCEs similarly indicates aberrant recombination, as this is a process frequently undertaken when the normal routes to error-free HRR are limiting (Al-Zain and Symington, 2021).

We identify the highly conserved P112 residue of NBS1, which lies within the short linker peptide between the N-terminal FHA and $BRCT_1$, as being required for the direct interaction with CYPA (Fig. 8). A regulated putative *cis-trans* isomerisation of the E111–P112 peptide bond could conceivably dynamically alter the relative positioning of the FHA domain with the tandem BRCTs of NBS1 (Fig. 7C,D). This may then impact on these domains' abilities to dynamically interact with their respective phospho-threonine (for FHA) and phospho-serine (BRCT) containing targets, consequently likely shaping/impacting NBS1 recruitment dynamics and/or plasticity of its interactome (Almawi Matthews and Guarné, 2017; Kim et al, 2021; Pennell et al, 2010). This is a hypothesis whose validation would require additional structural analysis using specialist techniques such as 2D-NMR which are outside the scope of this manuscript.

We found elevated and unresponsive HU-induced NBS1 and MDC1 foci formation in the absence of CYPA and following CYPA inhibition (Fig. 9). In addition, we found NBS1-P112G unable to undergo HU-induced NBS1 and RAD51 foci formation (Fig. 9D,E). It is tempting to speculate that these foci formation deficits may reflect the altered dynamism of FHA-BRCT target binding described above. It is also possible that this contributes to the impaired DNA replication (Fig. 2A,B), elevated fork stalling (Fig. 2C), HU-resistance (Fig. 11A) and dependence upon replication fork protection (Fig. 11C–E) we have catalogued in the absence of CYPA and following CYPA inhibition. Furthermore, we find a range of DNA replication and cell cycle factors in the CYPA-BioID (Fig. 5A; Appendix Table S1; Datasets EV1–3). Similarly, the CYPA-BioID throws up an extensive set of RNA binding factors (Fig. 4A,B; Appendix Table S2; Datasets EV1–3), many of whom may conceivably contribute to replication–transcription fork conflicts/collisions under conditions of CYPA-dysfunction. Untangling the functional hierarchy and significance of these putative CYPA interactions offer an exciting prospect and will involve significant further study.

PIN1, a well-known phosphorylation-directed PPI has been directly implicated in HRR-end resection; in one instance via interaction with CtIP (Steger et al, 2013; Zhou and Lu, 2016). PIN1

deficiency was found to increase resection and thus HRR in this study (Steger et al, 2013). Conversely, PIN1 activity was found to enhance BRCA1-BARD1 association with RAD51 thereby increasing RAD51 localisation at stalled replication forks (Daza-Martin et al, 2019). PIN1 has also been reported to stabilise BRCA1 (Luo et al, 2020). PIN1-dependencies are notoriously complex; simultaneously positively and negatively influencing processes in a context-dependent fashion (Zhou and Lu, 2016). Our data reveals a new means of resection regulation by a different PPI, the Cyclophilin family member CYPA, and we provide evidence that it directly interacts with the NBS1 component of the MRN complex (Figs. 7 and 8). We find that loss or inhibition of CYPA inhibits resection following replication fork stalling which consequently results in impairment of resection-driven DNA repair pathways, including SSA, but particularly HRR.

One area of CYPA biology that has attracted a lot of recent attention concerns its relationship with LLPS (liquid-liquid phase separation) of IDPs (intrinsically disordered proteins), particularly regulating IPDs such as α-synuclein and TDP-43, which are implicated in the neurodegenerative sequalae of disorders such as Amyotrophic Lateral Sclerosis (Babu et al, 2021; Babu et al, 2022; Favretto et al, 2020a; Favretto et al, 2020b; Lauranzano et al, 2015; Pasetto et al, 2021; Pasetto et al, 2017). In fact, it has been proposed that CYPA's role in controlling phase separation and proteome integrity may influence stem cell homoeostasis (Maneix et al, 2024). It's interesting to note the fundamental requirement of LLPS in mediating the chromatin recruitment of DNA repair proteins such as 53BP1, KU80, PARP1 and of specific relevance to our findings, recruitment of NBS1 (Levone et al, 2021). It is possible that CYPA may play a role in LLPS of key DNA repair proteins. Our CYPA interactome offers candidates worth exploring for this function.

In addition to describing a novel set of genome instability phenotypes associated with CYPA loss and inhibition, another tangible outcome of the CYPA interactome reported here is that it proposes multiple additional putative interactors. Alongside proteins involved in DNA repair, replication and transcription, we find several candidates involved in mitosis, kinetochores, centrosomes, and spindles collectively representing additional untrodden avenues to interrogate for new CYPA-dependent biology (Appendix Table S1). Whether any of the interacting candidates are direct prolyl isomerase or holdase-related targets of CYPA is worthy of pursuit. Our CYPA proximity interactome should serve as a useful resource for the community to seed additional studies, and in complementing another recently reported CYPA interactome (Maneix et al, 2024).

Cyclosporin A (CsA) is a long-established clinically important immunosuppressant used routinely since the early 1980s to prevent rejection and graft-versus-host disease in bone marrow and solid organ transplantation (e.g. kidney, heart and liver) (Fischer and Malesevic, 2013). CsA is a well-tolerated and widely used medicine, now also employed in dermatology for treating recalcitrant plague psoriasis, eczema, and atopic dermatitis, and even in ophthalmology for the management of dry eye disease. Our findings offer a logical mechanistic basis for the previously observed genotoxicity associated with prolonged exposure to high doses of CsA; pinpointing CYPA inhibition and its impact upon DNA repair specifically as a likely underlying contributor (IARC, 1990; O'Driscoll and Jeggo, 2008; Oliveira Zankl and Rath, 2004; Oztürk et al, 2008; Palanduz et al, 1999; Yuzawa et al, 1986). Interestingly,

CsA has shown efficacy against a range of different cancers including multiple myeloma, glioblastoma, retinoblastoma, gynaecological, hepatocellular, and non-small cell lung malignancies (Chan et al, 1996; Cohen et al, 2021; Eckstein et al, 2005; Lu et al, 2017; McLachlan et al, 1990; Morgan et al, 2007; Sood et al, 1999; Wang et al, 2017; Zhu et al, 2015). Whilst various processes have been invoked to explain these outcomes, such as CRK and prolactin receptor inhibition (Davra et al, 2020; Hakim et al, 2020), our findings now introduce impaired resection-driven DSB repair (incl. SSA and HRR) and associated genomic instability into this range of biological impacts. This is significant, as this CYPA-mediated HRR inhibition can be further exploited in a rational and targeted manner, as demonstrated here for MYCN-amplified NBm, MM and CML (Fig. 12; Dataset EV4).

Whilst an immunosuppressant would be undesirable as a routine cancer chemotherapy, multiple non-immunosuppressive CsA analogues (NIAs) have been developed as pan-genotypic hepatitis antivirals, and as treatments for increasingly common diseases such as fibrosis and non-alcoholic steatohepatitis (e.g. Alisporivir/Debio-025, NIM811, SCY-635, CRV431/Rencofilstat, Valspodar/PSC 833) (Daelemans et al, 2010; Flisiak and Parfienniuk-Kowerda, 2012; Han et al, 2022; Kuo et al, 2019; Peel and Scribner, 2015; Steadman et al, 2017; Trepanier Ure and Foster, 2017; Ure et al, 2020; Watashi, 2010). Our findings have potential clinical significance as they propose a feasible route to 'drugging' SSA and HRR through resection inhibition, potentially via the repurposing/repositioning of NIAs. This strategy could have efficacy in the context of certain cancers characterised by elevated RS coupled with an addiction to resection-driven DNA repair (i.e. elevated HRR) and/or impaired replication fork protection. Encouragingly, we find that at least one of these potent NIAs, namely NIM811 (N-methyl-4-isoleucine-cyclosporine; NCT00983060), robustly impairs DSB repair (Fig. 3D). It is even possible that the DNA repair phenotypes we've described here following CYPA PPI inhibition could contribute to the efficacy of NIAs in the context of inflammatory conditions such as fibrosis and non-alcoholic steatohepatitis, through the elimination of key relevant cell-types. Repurposing is a proven, rapid, and cost-effective strategy for widening therapeutic options for many diseases (Schein, 2020, 2021; Simsek et al, 2018; Strittmatter, 2014; Zhan Yu and Ouyang, 2022). Our findings provide an exciting route map for additional investigations with NIAs by leveraging their hitherto unknown impacts upon resection-driven DNA repair.

# Methods

## Cell culture

All lines were cultured in a humidified environment at 37 °C with 5% $CO_2$. Cell lines specifics are detailed in Appendix Materials and Methods Table S6: Cell lines.

## Compounds

Cyclosporin A ≥ 98.5% (HPLC) (Cat# 30024), Hydroxyurea (Cat# H8627), Nocodazole (Cat# SML1665) and Doxycycline hyclate (Cat# D9891) were obtained from Merck-SIGMA. NIM811 [CAS 143205-42-9] from MedChemexpress LLC was obtained via Insight Biotechnology LTD (Cat# HY-P0025).

## Chromosome spreads

AA8 Chinese Hamster Ovary cells were treated with 5 µM CsA and 1 µM nocodazole for 24 h prior to processing for chromosome spreads. Cells were swollen in 75 mM KCl for 10 min at 37 °C, fixed with Carnoy's fixative (methanol and glacial acetic acid at 3:1 ratio) for 10 min at room temp, before being applied dropwise to slide from over 30 cm. Slides were air-dried before staining with Giemsa (Merck-SIGMA Cat# 48900). For sister chromatic exchange analysis, cells were labelled with bromodeoxyuridine (BrdU 10 µM for 48 h) before treating with CsA (5 µM) and Colcemid (0.2 µg/ml) for ~12 h. Cells were swollen in KCl and fixed in Carnoy's fixative as above before being dropped on slides and air-dried overnight. The slides were treated with Hoescht (10 µg/ml) under light exclusion, washed in SSC buffer (2 M NaCl, 0.3 M tri-sodium citrate pH 7) before being UV irradiated (355 nm) for 1 h. Slides were counterstained with 6% Giemsa in Sorensen Buffer (1:1 mixture of 0.067 M $Na_2HPO_4$ and 0.067 M $KH_2PO_4$ pH 7.2), washed in $ddH_2O$ and dried overnight.

## CRISPR platform and plasmids

The *PPIA*/CYPA CRISPR system from *Origene* (Cat# KN203307) was used according to the manufacturer's instructions and is composed of KN203307G1; *PPIA* gRNA vector 1 in pCas-Guide vector (Target Sequence: GGCAATGTCGAAGAACACGG). KN203307G2; *PPIA* gRNA vector 2 in pCas-Guide vector (Target Sequence: GAA-CACGGTGGGGGTTGACCA). KN203307-D; donor DNA containing left and right homologous arms and GFP-puro functional cassette.

## Antibodies

These are listed as Appendix Materials and Methods Table S7: Antibodies.

## siRNA

Lipofectamine™ RNAiMAX Transfection Reagent (Thermo Fisher Scientific) was used according to the manufacturer's instructions. Briefly, cycling U2OS cells were seeded at a density of $3 \times 10^5$ cells per 6-cm² dish and transfected the same day with specific siRNA oligo pools (3–5× different oligos per gene target). Twenty-four hours post transfection, media was changed and the cells transfected for a 2nd time. 24 h after the 2nd transfection, media was again changed before pellets harvested after an additional 24 h. Protein expression was determined by semi-dry western blotting following SDS-PAGE of a denaturing extract (9 M Urea, 50 mM Tris-HCl pH 7.5, 10 mM 2-mercaptoethanol, sonicated at 20% amplitude for 10 s). A list of specific siRNAs is detailed in Appendix Materials and Methods Table S8: siRNA.

## Site-directed mutagenesis (SDM)

Target-specific SDM primers were designed using NEBaseChanger™ tool (https://nebasechanger.neb.com/) and mutagenesis undertaken using the Q5® Site-Directed Mutagenesis from (NEB UK). A list of target-specific oligos is detailed in Appendix Materials and Methods Table S9: Vectors and Site-Directed Mutagenesis (SDM).

## Survival and viability analyses

Clonogenic survivals were fixed and stained with 1% Methylene Blue in 20% methanol and colonies counted after 2 weeks. For Crystal Violet survival analyses $1 \times 10^5$ cells/well were seeded in six-well plates in 2 ml. After seven days, media was aspirated, and cells were fixed and stained with a 0.5% Crystal Violet (Merck-SIGMA, Cat# C6158) in 20% Methanol for 1 h. Plates were washed with $ddH_2O$ and left to dry. The stain was dissolved using 1 ml 10% Acetic Acid per well, diluted a further 1 in 20 in a cuvette and absorbance read at 595 nm. The 96-well format of the CellTitre Blue Viability Assay platform (Promega, Cat#G8080) for viability analyses following 72 h CsA treatment of suspension cultures of the MM and CML cell lines.

## 3D spheroid formation

A fibrinogen-based 3D culture approach was used to generate colonies from multiple myeloma (MM) patient-derived cell lines. Fibrinogen (Merck-SIGMA, Cat# F3879) was mixed with 2 M $CaCl_2$ and MM cells to form a 3D cross-linked jelly-like structure in 96-well plates. Seventy-two hours post CsA treatment, collagenase was added to the 3D culture to dissolve the matrix and viability determined using the CellTitre Blue Assay. Ultra-low attachment 96-well plates (Corning, Cat. #CLS3474) were used to generate tumour spheroids form the Neuroblastoma (NBm)-patient-derived cell lines following seeding at $1 \times 10^4$ cells/well.

## Flow cytometry

Cells were pulse labelled with 10 µM EdU (5-ethynyl-2'-deoxyuridine) for 30 min or treated with EdU + nocodazole (300 nM) for 24 h, then processed using Click-iT EdU Alexa Fluor-647 imaging kit according to the manufacturer's instructions (Thermo Fisher Scientific). Samples were counterstained with propidium iodide (5 µg/ml) supplemented with RNAase A (0.5 mg/ml), analysed using BD Accuri C6 Plus Flow Cytometer (BD Bioscience), and profiles processed using BD CSampler software (Becton Dickinson).

## DNA repair reporter GFP assay

The conventional cell lines used were EJ5-GFP-U2OS (for NHEJ), SA-GFP-U2OS (for SSA) and DR-GFP-U2OS (for HRR), but these were modified by Dr. Owen Wells (Genome Damage and Stability Centre, University of Sussex) using the *Sleeping Beauty* transposon system to stably integrate an I-SceI/Tet Doxycycline (Dox) inducible cassette to maximise GFP-repair signal. Cells were seeded at a density of $1 \times 10^5$ cells/well into a six-well plate and transfected with RNAiMAX or pre-treated with a drug. After 48 h 5 µg/mL Dox was added. Forty-eight hours following Dox treatment, cells were harvested, PBS washed and resuspended in 500 µL 3% BSA in PBS. Samples were subjected to flow cytometry using BD Accuri™ C6 Plus Flow Cytometer System (BD Bioscience). In total, $10^3$ events per sample were collected and GFP positive cells selected using BD CSampler software with background subtraction. Set Dox-induced GFP +ve cells as 100% and compare to those treated with drug or siRNA.

## DNA fibre combing and analyses

DNA fibres were prepared following combing from genomic DNA agarose plugs using the FiberComb Molecular Combing System (Genomic Vision) and processed for immunofluorescence detection according to the manufacturer's instructions. The antibodies used were mouse anti-BrdU (Becton Dickinson Cat# 347580) for IdU (Iododeoxyuridine) detection, rat anti-BrdU (Abcam Ab6326) for CldU (5-Chloro-2'-deoxyuridine) detection and rat anti-single-strand DNA (Genomic Vision). Secondary antibodies were Alexa Fluor (Invitrogen) anti-rat 488 (A21208), anti-mouse 594 (A31624) and anti-rabbit 647 (A31573). Images were captured and processed using the Olympus IX70 Fluorescence microscopy platform. A conversion factor of 2 kb/µm was used as is standard for combed fibres and fibre track lengths and fork speed data were processed using BoxPlotR (http://shiny.chemgrid.org/boxplotr/).

## BioID

We used the BioID1 version for proximity tagging via MYC-BirA-CYPA according to Roux K et al, 2012. The requisite plasmids were obtained from AddGene (https://www.addgene.org/); pcDNA3.1-MYC-BioID (Cat# 35700) and pcDNA3.1 MCS-BirA(R118G)-HA (Cat# 36047). The pcDNA3.1 mycBioID was a gift from Kyle Roux (Addgene plasmid # 35700; http://n2t.net/addgene:35700; RRID:Addgene_35700) and the pcDNA3.1 MCS-BirA(R118G)-HA was a gift from Kyle Roux (Addgene plasmid # 36047; http://n2t.net/addgene:36047; RRID:Addgene_36047). We also treated a population of cells with 1 mM HU for 18 h to compare with their untreated counterparts. Nuclear extracts were resolved on an 8% SDS-PAGE (resolution range 55–250 KDa) and a 15% SDS-PAGE (for a resolution range of 15–55 KDa). Lanes were cut into a total of 12× equally sized slices and then sent to *FingerPrint Proteomics Facility* at the University of Dundee for processing and mass spectroscopy analysis (https://www.dundee.ac.uk/locations/fingerprints-proteomics-facility).

## Cell extract and fractionation

Whole-cell extracts were prepared using urea lysis buffer (9 M urea, 50 mM Tris-HCL at pH 7.5, 10 mM β-mercaptoethanol), followed by sonication at 30% amplitude for 15 s. For Nuclear and Cytoplasmic fractions, cells were resuspended in ice-cold NP40 buffer (0.1% NP40/IGEPAL in PBS, 1x protease inhibitor *cOmplete$^{TM}$* cocktail (Roche)). This was centrifuged for 30 s at $10^3$ RPM at 4 °C and the top portion of the supernatant was removed into a fresh tube (cytoplasmic fraction). The remaining supernatant was removed and the nuclear pellet washed in ice-cold NP40 buffer, the nuclear pellet was then subjected to the urea lysis buffer method. The protein concentration of each of the fractions was determined (Quick Start™ Bradford Protein Assay, (Bio-Rad)) and suspended in 5× Laemmli Buffer, then boiled prior to western blot analysis.

## Immunoprecipitation

For transient ectopic overexpression work, HEK293 cells were transfected using the calcium phosphate method (2 M CaCl$_2$ with HEPES-buffered saline: 50 mM HEPES pH 7, 280 mM NaCl,

1.5 mM Na$_2$HPO$_4$) with a media change at 24 h and before harvesting at 48 h. A non-denaturing cell extract was prepared by incubating pellets on ice for 1 h with 200 µl-1 ml IP buffer (50 mM Tris-Cl pH 7.5, 150 mM NaCl, 2 mM EDTA, 2 mM EGTA, 25 mM NaF, 25 mM β-glycerophosphase, 0.1 mM sodium orthovanadate, 0.2% Triton X-100, 0.3% NP40, 1× protease inhibitor *cOmplete$^{TM}$* cocktail (Roche) supplemented with benzonase at 1:1000). Samples were pelleted and the supernatant soluble cell extract used for immunoprecipitation (IP). Magnetic Dynabeads™ Protein G (Invitrogen) were used for IP of endogenous proteins. ANTI-FLAG® M2 Affinity Gel Beads (Merck-SIGMA) were used to IP FLAG-tagged protein, which were eluted FLAG® peptide (Merck-SIGMA) according to the manufacturer's instructions.

## Recombinant protein production

Gene block (gBlock) fragments were obtained from *Integrated DNA Technologies (IDT)* after being codon optimised for *E. coli* using *IDT* Codon Optimization Tool and designed with the restriction sites Nco1 and Xho1 on each end. Backbone vector pET-15b (69661-3) and gBlocks were digested with NEB CutSmart buffer (Cat# B72045) using restriction enzymes Nco1-HF (high fidelity Cat# R3193S) and Xho1- Cat# R0146S both from *NEB*. DNA was run on 2% agarose gel and extracted using QIAquick Gel Extraction Kit (*QIAGEN*). Fragments and backbone were ligated using T4 DNA ligase (Cat# M0202) from *NEB*. *E. coli* strain BL21(DE3) cells (*NEB*) were transformed and initially streaked to plates before a single colony was picked for a 10 ml starter culture with selection and grown overnight. Saturated starter culture was expanded to 500 ml and left to grow at 37 °C. Once OD 600 was between 0.6 and 0.8, cells were induced with 1 mM IPTG @ 18 °C overnight. Cells were pelleted @ 4 °C before the dry pellet was snap-frozen. Frozen pellets were thawed on ice and lysed with 20 ml buffer per gram of pellet (50 mM HEPES, 500 mM NaCl, 0.5 mM TCEP, 1U/ml DNase supplemented with *Roche cOmplete™*, EDTA-free Protease Inhibitor Cocktail). The lysate was sonicated on ice at 40% amplitude, 5 s on and 10 s off, for a total sonication time 5–10 min. Samples were centrifuged at 20 K RPM (Beckman Coulter SLA-3000) for 1 h at 4 °C and the supernatant was filtered using 0.5 µM filter. To purify HIS-tagged proteins, HiTrap® TALON® Crude (Cytiva, Cat# 28-9537-66) was used and Streptavidin-tagged proteins were purified using Strep-Tactin® Superflow® Plus cartridge (*QIAGEN*). Columns were attached to MINIPULS® 3 peristaltic pump (*Gilson*) and washed with 3× column volume ddH2O and equilibrated with 5× column volume lysis buffer. For HIS-purifications, 5 mM imidazole was added to the protein lysate before running it through the column. Column was then washed with 10× volume of buffer containing 10 mM imidazole. Protein was eluted with 300 mM imidazole. For Streptavidin purifications the same process was undertaken but without imidazole. Streptavidin-tagged proteins were eluted with 2 mM D-desthiobiotin using the peristaltic pump, as per the manufacturer's instructions. PD10 desalting columns (*Cytiva*) were used to remove salts from purified protein and to exchange to freezing buffer (20 mM HEPES, 100 mM NaCl, 0.5 mM EDTA, TCEP 0.5 mM and *Roche cOmplete™* Protease Inhibitor Cocktail), using the gravity protocol. Glycerol was slowly added to final concentration of 5% before protein was aliquoted to single use, snap-frozen and stored at −80 °C.

## Recombinant protein interaction

Recombinant Strep-tagged CYPA and HIS-tagged FHA-BRCT$_1$ protein were combined and then rotated for 30 min 4 °C. Meanwhile MagStrep "type3" XT Beads (IBA Lifesciences GmbH) were washed and equilibrated with pulldown buffer (100 mM Tris-Cl pH 8, 150 mM NaCl, 1 mM EDTA, 0.5 mg/ml BSA). 50 µl of beads were added to each pulldown and incubated for 1 h, with rotation at 4 °C. To control for non-specific binding to the beads, HIS-tagged protein was also incubated with beads alone. Beads were washed with buffer 3x using a magnetic rack. To elute protein, pulldown buffer supplemented with 50 mM biotin (IBA Lifesciences GmbH) formulated in elution buffer (100 mM Tris-Cl pH 8, 150 mM NaCl, 1 mM EDTA) was added to the beads for 30 min, with rotation at 4 °C.

## Microscopy

Indirect immunofluorescence (IF) images were captured using the *Olympus* IX71 fluorescence microscope with a 60x objective using Micro-Manager software (https://micro-manager.org/) and the foci data processed using Cell Profiler software (https://cellprofiler.org/). For CYPA IF following pre-extraction, slides were exposed to 0.2% ice-cold Triton X-100 in PBS for 30 s prior to fixation. Slides were fixed in 3% PFA with 2% sucrose for 10 min and room temp. For foci formation following transfection into NBS-ILB1, cells seeded onto coverslips were transfected using Lipofectamine™ 3000 following the manufacturer's instructions (Thermo Fisher Scientific Inc.). Media was changed after 24 h. Forty-eight hours post transfection, cells were treated with 2 mM HU for 3 h before being fixed with 3% PFA/2% sucrose in PBS. For NBS1 foci slides were pre-extracted with 0.2% ice-cold Triton X-100 in PBS, washed three times in PBS prior to fixation.

## Bioinformatics and structural software

For gene ontology analysis, we used the Gene Ontology Resource (http://geneontology.org/). UniProt IDs were converted to gene symbols using the SYNGO platform (https://www.syngoportal.org/convert). We used the MUSCLE tool for gene alignment (https://www.ebi.ac.uk/Tools/msa/muscle/). Gene variant analyses were undertaken using PolyPhen 2 (http://genetics.bwh.harvard.edu/pph2/) and MutationTaster (https://www.mutationtaster.org/), whilst variant cataloguing was undertaken using gnomAD Browser (https://gnomad.broadinstitute.org/), cBioPortal TCGA pan-cancer Atlas Studies dataset (https://www.cbioportal.org/) and the COSMIC (Catalogue of Somatic Mutations in Cancer) portal (https://cancer.sanger.ac.uk/cosmic). The structure of human NBS1 (UniProt: O60934) was extracted from the AlphaFold Protein Structure Database (https://alphafold.ebi.ac.uk/), Jumper J et al, 2021 and Varadi M et al, 2022. Peptide binding modelling of human Cyclophilin A to the NBS1 FHA-BRCT$_1$ linker peptide was generated using the CoLabFold v1.5.2-patch: AlphaFold2 using MMseqs2 (https://colab.research.google.com/github/sokrypton/ColabFold/blob/main/AlphaFold2.ipynb#scrollTo=kOblAo-xetgx), Mirdita M et al, 2022. *PPIA* CRISPR knockout Fitness Score data were obtained from the DepMap cancer dependency map *Project Score* portal (https://score.depmap.sanger.ac.uk/). *BRCA2* status of

individual breast carcinoma cell lines was extracted from the *Cell Model Passports* portal (https://cellmodelpassports.sanger.ac.uk/). Protein disorder analyses of NBS1 FHA-Linker-BRCT$_1$ peptides was undertaken using the PONDR (Prediction Of Natural Disordered Regions) tool (http://www.pondr.com/), ANCHOR2 and IUPred3 analysis using the IUPred3 web interface (https://iupred3.elte.hu/) and context-dependent protein binding behaviour using the FuzzPred platform (https://fuzzpred.bio.unipd.it/predictor).

## Data availability

No data has been deposited in any public database. All Source Data is available via download from the respective figure.

The source data of this paper are collected in the following database record: biostudies:S-SCDT-10_1038-S44319-024-00184-9.

## Peer review information

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

## Acknowledgements

This work was funded by the UKRI Medical Research Council grant MR/T012978/1. Special thanks to Dr. Owen Wells (Genome Damage & Stability Centre, Uni of Sussex, UK) for creating and providing the modified DNA repair GFP-reporter cell lines. Thanks to Dr. Lesley Hart (Genome Damage & Stability Centre, Uni of Sussex, UK) for CML-related work. Thanks to Prof. Madalena Tarsounas (Uni of Oxford, UK) for the conditional shBRCA2 cell line, and thanks also to Dr. Antony Oliver (Genome Damage & Stability Centre, Uni of Sussex, UK) for advice regarding recombinant protein expression and for structure-related discussion and insights.

## Author contributions

**Marisa Bedir**: Data curation; Formal analysis; Investigation; Methodology; Writing—review and editing. **Emily Outwin**: Data curation; Writing—review and editing. **Rita Colnaghi**: Data curation; Writing—review and editing. **Lydia Bassett**: Data curation; Writing—review and editing. **Iga Abramowicz**: Data curation; Writing—review and editing. **Mark O'Driscoll**: Conceptualisation; Resources; Formal analysis; Supervision; Funding acquisition; Investigation; Writing—original draft; Project administration; Writing—review and editing.

Source data underlying figure panels in this paper may have individual authorship assigned. Where available, figure panel/source data authorship is listed in the following database record: biostudies:S-SCDT-10_1038-S44319-024-00184-9.

## Disclosure and competing interests statement

The authors declare no competing interests.

