## [Peer Review File · EMBO Reports]

A novel role for Cyclophilin A in DNA-repair via the MRE11-RAD50-NBS1 complex.

Marisa Bedir, Emily Outwin, Rita Colnaghi, Lydia Bassett, Iga Abramowicz, and Mark O'Driscoll

Corresponding author(s): Mark O'Driscoll (M.O-Driscoll@sussex.ac.uk)

Review Timeline:

Transfer Date:	2nd Feb 24
Editorial Decision:	9th Feb 24
Appeal Received:	4th Apr 24
Editorial Decision:	16th May 24
Preliminary Response to Reviewers:	17th May 24
Revision Received:	28th May 24
Accepted:	5th Jun 24

Editor: Esther Schnapp

**Transaction Report: This manuscript was transferred to
EMBO reports following peer review at Review Commons.**

Review
COMMONS

Review #1

1. Evidence, reproducibility and clarity:

Evidence, reproducibility and clarity (Required)

In this manuscript, the authors reveal a previously unexplored role of CYPA in DNA repair, particularly in the context of cells sensitive to the CsA. The authors' multi-faceted approach involved using CRISPR/Cas9-engineering, siRNA, BioID, co-immunoprecipitation, and specific DNA repair investigations. They suggest that CYPA, through its PPI activity, plays an active role in DNA repair, specifically in DNA end resection. They also demonstrate that inhibition or loss of CYPA results in impaired HRR following DNA replication fork stalling. Furthermore, the authors associate the loss and inhibition of CYPA with certain genetic vulnerabilities, suggesting potential therapeutic applications by exploiting CYPA PPI inhibition to selectively target cancer cells with characteristic genomic instability.

The manuscript presents clear and comprehensive data, demonstrating the profound impact of CYPA on DNA repair. It would be suitable for publication after addressing the following points:

1. It's surprising to find that the loss of CYPA abolished HU-induced NBS1 foci, as the MRE11 interactive domain of NBS1 should remain intact in CYPA deficient conditions and the N-terminus of NBS1 is dispensable for ATM activation (Kim et al., 2017; Stracker and Petrini, 2011). A more detailed mechanistic explanation of this phenotype would be appreciated. The authors should check the subcellular localization of NBS1 and the stability of MRN in wildtype and CYPA KO cells. Additionally, including the kinetics of NBS1 foci formation using multiple timepoints in wildtype and CYPA KO cells after damage will further support the observation.
2. The authors showed that the interaction between CYPA and MRN didn't change after HU treatment. The authors should also include co-localization analysis of CYPA and NBS1 after HU.
3. The paper demonstrated that BRCA2 knockdown cells were sensitive to CsA. The authors should also examine CsA sensitivity in BRCA2 deficient cancer cells. In addition, the authors could elaborate more on their criteria for selecting cancers for CYPA inhibition, whether it is based on high genomic instability or an addiction to HRR for survival.

2. Significance:

Significance (Required)

The significance of this study is twofold: it adds a new layer to our understanding of DNA repair mechanisms and, importantly, it could point the way to novel therapeutic strategies for cancer. It will spark interest from molecular biologists to clinicians and pharmaceutical researchers.

3. How much time do you estimate the authors will need to complete the suggested revisions:

Estimated time to Complete Revisions (Required)

(Decision Recommendation)

Between 1 and 3 months

No

Review #2

1. Evidence, reproducibility and clarity:

Evidence, reproducibility and clarity (Required)

The current manuscript by Bedir et al. explores the role of cyclophilin A in DNA repair, particularly homologous recombination. Authors show that the absence of cyclophilin A or loss of PP1 activity affects end-resection via direct interaction with

NBS1. Authors have conducted a series of experiments to confirm their findings. While the findings are interesting, further discussion/ experiments mentioned below will perhaps assure readers with respect to pointed direct vs consequence facilitated indirectly through global cellular effects of CYPA.

1. Authors show delayed S-phase transit along with reduced replication speed indicating replication stall. However, authors have not discussed how cyclophilinA might regulate replication (other than hypothesizing regarding altered dynamism of FHA-BRCT). It is conceivable that it could be an indirect effect on cellular metabolism or if authors believe it could be due to direct disruption to core replication machinery or signaling. In this regard, it will be helpful to see if there is shortening of (premature entry) G1 phase and comment on the status of the associated G1/S checkpoint.
2. In connection to this, it will also be interesting to see if the ATR/Chk1 signaling axis is intact in CYPA KO cells with or without additional DNA damage compared to WT.
3. Authors show that the P112 residue of NBS1 is important for the binding of cyclophilinA.
4. What is the status of interaction among components of the MRN complex in CYPKO cells and P112G NBS1? Further, what are the authors' thoughts on rescue experiments and whether P112G containing NBS1 to perform resection function.
5. What are the protein levels of MRN, RAD51 etc. in CYPKO cells? It will be important control to delineate the effects of CYPA on global transcription and translation vs specific and direct effect on end-resection. Can overexpression of NBS1 rescue the observed resection and focus phenotypes?

2. Significance:

Significance (Required)

Current study highlights the role of cyclophilin A or in large peptidyl-prolyl cis- trans isomerases activity in DNA repair. Although this is not the first study showing the relevance of cyclophilin A in DNA repair, they do highlight its role in homologous recombination and DNA repair. Authors have quite conclusively explored the interaction between NBS1 and cyclophilinA as well as the putative proline residue important for this interaction.

One of the drawbacks of the study is the pleiotrophic effects of CyclophilinA. This needs to be at least discussed. Authors themselves observe induction of DSBs, replication stall, reduced NHEJ, SSA as well as HR efficiencies. Taken together, the effects of CyclophilinA even on resection could be a combination of both direct and

indirect effects.

This manuscript will have broad interest from groups working on genomic stability, immunology as well as cancer therapy. I have expertise in NHEJ, mammalian replication and replication-stress response.

3. How much time do you estimate the authors will need to complete the suggested revisions:

Estimated time to Complete Revisions (Required)

(Decision Recommendation)

Between 1 and 3 months

Yes

Review #3

1. Evidence, reproducibility and clarity:

Evidence, reproducibility and clarity (Required)

In this manuscript, O'Driscoll and colleagues identify the role of cyclophilin A (CYPA), a peptidyl-prolyl cis-trans isomerase, in promoting DNA repair. They propose that the catalytic activity of CYPA is required for the action of the MRE11-RAD50-NBS1 (MRN) complex and thus double-strand break (DSB) repair. This

study originated from their previous finding that cyclosporin A (CsA) induces replication-associated DNA breakage and genome instability in LIG4 syndrome patient fibroblasts. As CsA is an inhibitor of the CYP A, the authors reasoned that the negative effect of CsA in DNA repair results from its inhibition of CYP A and presumably its essential downstream substrates in DNA repair. Using CRISPR/Cas-based U2OS knockout cells, they showed that the catalytic activity of CYP A is necessary for homology-directed repair. Series of BioID-proximity interactome analysis, biochemical studies (e.g., co-immunoprecipitation), and AlphaFold-derived structural determination revealed that CYP A directly interacts with the MRN complex, specifically through the Pro112 of NBS1, and its catalytic activity is required for damage-induced NBS1 foci formation, which all together led to the conclusion that the MRN complex is a direct substrate of CYP A and that CYP A controls DNA end resection and DSB repair via isomerization of NBS1.

This is an interesting story as it reveals a new role of prolyl isomerization, which is mediated by CYP A, in promoting DSB repair. The identification of NBS1 as a novel substrate of CYP A is significant, and the manuscript may provide new insight into how prolyl isomerization of NBS1 regulates the function of the MRN complex that is engaged in DNA end resection during DSB repair. However, the authors' major claim that CYP A controls DSB repair via the MRN complex is not substantiated by the data provided at its current form. Either evidence is lacking or experiments were not performed in a convincing way. Their conclusion should be supported by additional key experiments to prove that the catalytic activity of CYP A is indeed required for DSB repair and NBS1 is a major substrate of CYP A, through which CYP A regulates DNA end resection at the stalled DNA replication fork.

****Major comments****

1. The authors reconstituted the CYP A knockout (KO) cells with WT or a catalytic mutant (R55A) for the structure-function analysis. However, re-expression levels of CYP A are vastly different between WT vs. R55A, R55A being expressed at much lower levels (not near to endogenous CYP A) (Fig. 1C). Consequently, the loss-of-function phenotypes of R55A may be simply explained by its inadequate reconstitution, thus failing to complement KO phenotypes. For instance, the lack of pRPA2 S4/S8 induction in R55A cells may be just due to the insufficient expression of R55A, thus resulting in the same phenotype as KO. Additionally, the R55A cells were compared to parental cells, not to the WT-reconstituted cells for the majority of functional analysis, so it is not clear whether WT is able to complement the KO phenotype in their system (Figs. 1, 2, 6, 9, 10, and 11). Whether the catalytic activity of CYP A is indeed responsible for the phenotypes of DNA repair deficiency is not supported. The authors should compare the phenotypes between WT- vs. R55A-

reconstituted cells side-by-side for the key experiments. Ideally, expression of WT and R55A should be similar in KO cells to exclude the possibility that the R55A phenotypes merely result from insufficient mutant expression rather than true loss of catalytic activity.

2. Another major concern is that the evidence to support that NBS1 is the major substrate of CYPA is lacking since all the experiments were performed with the CYPA mutant or CsA treatment. Whether the NBS1 P112G isomerization-defective mutant indeed exhibits a defect in DNA repair similarly to the CYPA mutant is not shown. For instance, one key experiment would be to test whether the P112G mutant fails to form damage-inducible NBS1 foci formation.

3. In Figure 7A, the authors showed that the interaction between CYPA and NBS1 is dependent on the isomerization activity of CYPA. It should be checked whether the CYPA R55A mutant fails to interact with NBS1 in contrast to WT to support the main conclusion that NBS1 is controlled by the isomerization activity of CYPA.

4. OPTIONAL) One major weakness of this study is that it focuses on characterizing the interaction between CYPA and NBS1, then jumps into a conclusion that the catalytic activity of CYPA is required for DSB repair based on its direct interaction with NBS1. How the isomerization of NBS1 affects its localization, stability, and/or function is not addressed. At its current form, the functional link between NBS1 isomerization and stalled fork processing is weak. Elucidating how the catalytic activity of CYPA controls the action of the MRN complex via the isomerization of NBS1 will add significant impact on the manuscript. Otherwise, the story fails to fully support the description of its title.

****Minor comments****

1. Fig. 1E; is the survival between KO and R55A statistically significant? If so, do the authors have an explanation? Why is the reconstitution of R55A more toxic than KO alone?

2. In Fig. 3D, the NHEJ activity of CsA- or NIM811-treated cells is significantly downregulated in comparison to control, which raises the possibility of the pleiotropic effect of CYPA inhibition. The authors should discuss this issue.

3. In Figure 8A, since the expressions of Flag-NBS1 WT, P112G, and P64G are very different, the conclusion that the isomerization of CYPA is essential for NBS1 cannot be supported. Given the variation of input levels, it appears that the P64G mutation actually enhances the interaction with endogenous CYPA. Is this reproducible? This co-IP result may need to be quantified from independent sets for statistical analysis.

4. A defect in DSB repair generally hypersensitizes cells to DNA replication stress, including HU. In this regard, resistance of the CYPA KO (or R55A cells) to HU is interesting, but it may be due to the nonspecific effect of the CYPA loss in multiple DNA damage signaling and repair processes. Alternatively, cell cycle may be affected

nonspecifically, rendering cells resistant to replication-associated genotoxic stress. This needs to be addressed further. Analysis of overall cell cycle profile may be required.

5. Text not to mention Abstract is too dense. The manuscript will benefit a lot from extensive editing and rearrangement of figures to make the story more succinct for journal submission.

****Referees cross-commenting****

I agree with concerns on the pleiotropic effect of CYPA KO, which exhibit many distinct phenotypes in DNA repair and replication fork stability.

2. Significance:

Significance (Required)

While establishing a new link between CYPA-dependent prolyl isomerization and DSB repair is significant, the current manuscript suffers lack of evidence to support its main conclusion. Specifically, although the role of CYPA in DNA repair is fairly well described using its inhibitor or KO cells, whether its isomerization activity is indeed essential and whether NBS1 is the major target for its action in DSB repair is not clear. Existence of many other targets cannot be excluded. Whether the role of CYPA is specific to replication-associated DSB repair processes or can be generally applicable to homologous recombination in any DSB repair is not shown. The role of MRN in stalled fork processing and in response to DSBs could be different, but how CYPA would modulate these distinct processes is not addressed. As such, the manuscript is targeting more specialized audience, but if the link between CYPA and the MRN complex can be further elaborated, including how isomerization affects the function of NBS1 (e.g., using the isomerization-defective NBS1 mutant), it could reach out to broader readership.

My field of expertise includes DNA replication stress and replication-associated repair processes including stalled fork processing and recovery. I am familiar with most of the genetic, cellular, and biochemical experiments presented in the manuscript. I do not have significant expertise on the structural analysis of protein-protein interactions.

3. How much time do you estimate the authors will need to complete the suggested revisions:

Estimated time to Complete Revisions (Required)

(Decision Recommendation)

Cannot tell / Not applicable

No

Full Revision

Manuscript number: RC-2023-02068

Corresponding author(s): Prof. Mark O'Driscoll

A novel role for the peptidyl-prolyl *cis-trans* isomerase Cyclophilin A in DNA-repair following replication fork stalling via the MRE11-RAD50-NBS1 complex.

Bedir M *et al.* BioRxiv 2023-546694v1. <https://doi.org/10.1101/2023.06.27.546694>

1. General Statements [optional]

Dear editorial team,

We are excited to offer this revised manuscript for consideration as the first multifaceted and comprehensive demonstration of a novel link between the peptidyl prolyl *cis-trans* isomerase Cyclophilin A (CYPA) and the DNA-R/genome stability network. In this paper we showcase a CYPA-BioID generated interactome, identifying a plethora of novel putative CYPA interactors. We validate several interactions with well-characterized DNA-R proteins and show that CYPA can directly interact with NBS1. We document a range of DNA-R deficits caused by loss and/or inhibition of CYPA and identify a series of CYPA dependencies/vulnerabilities within the genome stability network. Collectively, these represent completely new biological insights into the consequences of impaired CYPA function. Finally, we demonstrate the targeted application of CYPA inhibition can be effective in a set of cancers sharing a genomic profile of elevated replication stress and homologous recombination dependency/addiction. This is significant, as several clinically validated CYPA inhibitors already exist (e.g., Alisporivir, NIM811, SCY-635, Rencofilstat). Whilst these non-immunosuppressive Cyclosporin A analogues are currently being trialed mainly as anti-virals (pan-genotypic hepatitis, HIV, SARS-CoV2), we provide evidence that strongly argues that their repurposing/repositioning against select cancers should be considered.

We believe our findings are of interest to a wide readership, a point best illustrated by Reviewer #1's comment: "*It will spark interest from molecular biologists to clinicians and pharmaceutical researchers*". Furthermore, considering we present a comprehensive CYPA interactome thereby proposing a wealth of additional research avenues worth pursuing and claim primacy for linking CYPA function/dysfunction to DNA-R/genomic instability, we think the wider impact and future citation potential of this paper are extremely high.

We are very grateful to **Review Commons** for organizing the rapid peer review of the preprint version of this manuscript. Largely excepting Reviewer #3, we appreciate the Reviewers' constructive evaluations and individual feedback. As corresponding author, originator, and director of this CYPA research, I do take some issue with the approach, condescending tone, and overly dismissive review from Reviewer #3, which runs contrary to my understanding of **Review Commons'** intent. As a referee myself of ~25yrs experience it's

Full Revision

being that certain characters take this approach. Setting this review alongside the other Reviewer comments expose it for what it is.

We have revised the manuscript, select figures, table and supplementary material according to various Reviewer comments and requests. We also include additional new data. This is detailed in the point-by-point response to the Reviewers.

All the best

Mark O'Driscoll
(on behalf of the authors).

This section is mandatory. Please insert a point-by-point reply describing the revisions that were already carried out and included in the transferred manuscript.

Revision summary.

Additional new data.

- CYPA expression levels in Scrm Vs KO Vs R55A isogenic cell lines as new **Fig 1C**.
- ATR signaling: western blot analysis of HU-induced p-CHK1 (S345) in Scrm, KO and R55A isogenic cell lines as new **Suppl Fig 1B**.
- MRN expression: western blot analysis of expression of NBS1, MRE11, RAD50 and MCM2 in Scrm, KO and R55A isogenic cell lines as new **Suppl Fig 7A**.
- NBS1 subcellular fractionation: western blot analysis of NBS1 from whole cell extract Vs cytoplasmic extract Vs nuclear extract comparing expression/distribution in Scrm, KO and R55A isogenic cell lines, as new **Suppl Fig 7B**.
- CYPA immunofluorescence (IF) staining on untreated and HU treated U2OS, as new **Suppl Fig 7C**.
- CYPA immunofluorescence (IF) staining on untreated and HU treated U2OS following pre-extraction, as new **Suppl Fig 7D**.
- DepMap *Project Score Cancer Gene Dependency* cell survival ("fitness") following *PPIA/CYPA-KO* in breast carcinoma cell lines mapped against BRCA2 status, as a new **Suppl Table 5**.
- DepMap *Project Score Cancer Gene Dependency* cell fitness following *PPIA/CYPA-KO* in Neuroblastoma cell lines, as a new **Suppl Spreadsheet 4**.
- DepMap *Project Score Cancer Gene Dependency* cell fitness following *PPIA/CYPA-KO* in Multiple Myeloma cell lines, as a new **Suppl Spreadsheet 4**.

Full Revision

- DepMap *Project Score Cancer Gene Dependency* cell fitness following *PPIA/CYPA-KO* in Chronic Myelogenous Leukaemia cell lines, as a new **Suppl Spreadsheet 4**.

Revised and/or additional text.

The *Abstract, Introduction, Materials & Methods, Results* and *Discussion* have been amended as necessary, to facilitate the issues raised by the Reviewers.

Point-by-point response to Reviewers.

Reviewer #1.

We thank this reviewer for their understanding and appreciation of our CYPA study as espoused by their comprehensive summary of the content, importance, and potential implications of our work; *“The manuscript presents clear and comprehensive data, demonstrating the profound impact of CYPA on DNA repair.”* Furthermore, we very much appreciate their robust and complementary words regarding the significance of our work and its wide appeal; *“The significance of this study is twofold: it adds a new layer to our understanding of DNA repair mechanisms and, importantly, it could point the way to novel therapeutic strategies for cancer. It will spark interest from molecular biologists to clinicians and pharmaceutical researchers.”*

Query:

It's surprising to find that the loss of CYPA abolished HU-induced NBS1 foci, as the MRE11 interactive domain of NBS1 should remain intact in CYPA deficient conditions and the N-terminus of NBS1 is dispensable for ATM activation (Kim et al., 2017; Stracker and Petrini, 2011). A more detailed mechanistic explanation of this phenotype would be appreciated. The authors should check the subcellular localization of NBS1 and the stability of MRN in wildtype and CYPA KO cells. Additionally, including the kinetics of NBS1 foci formation using multiple timepoints in wildtype and CYPA KO cells after damage will further support the observation.

RESPONSE:

Regarding NBS1 foci formation, we note that rather than *abolish* HU-induced NBS1 foci formation, CYPA loss (through KO) and/or inhibition (through p.R55A) in fact results in a *“...spontaneously elevated yet unresponsive amount of NBS1 foci/cells when compared to scrambled”* (see original **Fig 9A** legend and associated Results section text). We have reinforced this observation in the revised Results section entitled *‘CYPA influences NBS1 and MDC1 foci formation’* and in the *Discussion* section. We do describe a kinetic impairment of

RAD51 foci formation in the CYPA-engineered lines up to 16hrs post HU-treatment (**Fig 6D**). Our mechanistic working model is that CYPA interacts directly with NBS1 via a Pro residue within the short linking peptide between the FHA and BRCT₁, and that this likely influences the relative dynamic positioning of the FHA with BRCA₁-BRCT₂, at least following acute HU treatment; replication fork stalling, likely biased towards ATR-dependent signaling initially, rather than that of ATM. The relative positioning of these functional domains can impact MRN function, and we discuss this possible mechanism in the section entitled '*CYPA and the MRN complex*', with reference to the detailed structure-function analyses and complementary DDR activation models described by

- Williams, R.S., et al., *Nbs1 flexibly tethers Ctp1 and Mre11-Rad50 to coordinate DNA double-strand break processing and repair*. Cell, 2009. 139(1): p. 87-99.
and
- Lloyd, J., et al., *A supramodular FHA/BRCT-repeat architecture mediates Nbs1 adaptor function in response to DNA damage*. Cell, 2009. 139(1): p. 100-11.
and
- Rotheneder, M., et al., *Cryo-EM structure of the Mre11-Rad50-Nbs1 complex reveals the molecular mechanism of scaffolding functions*. Mol Cell, 2023. 83(2): p. 167-185.e9.

The N-terminal FHA-BRCT region of NBS1 does indeed influence MRN recruitment and HRR execution, a point we highlight in the section entitled '*CYPA influences NBS1 and MDC1 foci formation*', with reference to the seminal original observations of

- Sakamoto, S., et al., *Homologous recombination repair is regulated by domains at the N- and C-terminus of NBS1 and is dissociated with ATM functions*. Oncogene, 2007. 26(41): p.6002-6009
and
- Tauchi, H., et al., *The forkhead-associated domain of NBS1 is essential for nuclear foci formation after irradiation but not essential for hRAD50-hMRE11-NBS1 complex DNA repair activity*. J Biol Chem, 2001. 276(1): p. 12-15.
and
- Zhao, S., W. Renthal, and E.Y. Lee, *Functional analysis of FHA and BRCT domains of NBS1 in chromatin association and DNA damage responses*. Nucleic Acids Res, 2002. 30(22): p. 4815-22.
and
- Cerosaletti, K.M. and P. Concannon, *Nibrin forkhead-associated domain and breast cancer C-terminal domain are both required for nuclear focus formation and phosphorylation*. J Biol Chem, 2003. 278(24): p. 21944-21951.

HU-unresponsive NBS foci (indicative of MRN dysfunction) and MDC1 foci formation are consistent with the DNA-R (i.e., DR-GFP reporter systems: **Fig 3A-C** and impaired RAD51 foci formation: **Fig 6D**) and resection-related phenotypes (**Fig 6A-B**) we report here and are also consistent with the relative resistance to HU-induced killing we report for CYPA-KO and CYPA-R55A cells (**Fig 11A** and as reported by Manthey, K.C., et al., *NBS1 mediates ATR-dependent RPA hyperphosphorylation following replication-fork stall and collapse*. J Cell Sci, 2007. **120**(Pt 23): p. 4221-9).

Full Revision

At the reviewer's request we include additional novel experimental data showing that MRN expression is stable and equivalent in control, CYPA-KO and CYPA-R55A cells (**Suppl Fig 7A**). We also provide evidence that NBS1 subcellular distribution (via extract fractionation) is not altered upon CYPA loss and/or inhibition (**Suppl Fig 7B**).

Query:

The authors showed that the interaction between CYPA and MRN didn't change after HU treatment. The authors should also include co-localization analysis of CYPA and NBS1 after HU.

RESPONSE:

At the reviewer's suggestion we undertook a series of IF analyses concerning endogenous CYPA (i.e., +/- HU, +/- pre-extraction). We found that endogenous CYPA failed to form foci following HU thereby precluding CYPA-NBS1 foci co-localization analysis (**Suppl Fig 7C-D**).

Query:

The paper demonstrated that BRCA2 knockdown cells were sensitive to CsA. The authors should also examine CsA sensitivity in BRCA2 deficient cancer cells. In addition, the authors could elaborate more on their criteria for selecting cancers for CYPA inhibition, whether it is based on high genomic instability or an addiction to HRR for survival.

RESPONSE:

Despite repeated attempts we have been unable to successfully routinely culture the TNBC suspension line HCC1599 (*BRCA2* c.4154_5572del1419 and p.K1517fs*23), consistent with its reported ~5 days population doubling time. Although not a tumour line *per se*, we also failed to effectively culture the FANC-D1 patient FB line HSC62 (*BRCA2* c.8488-1 G>A (IVS19-1G>A)) to enable survival analysis. We provide new quantification analysis of the CsA survival on the H1299 conditional sh*BRCA2* line (**Fig 11E**). Additionally, we include a comprehensive new analysis of cell survival ("fitness") of a range of breast carcinoma cell lines following *PPIA*/*CYPA*-KO, extracted from DepMap *Project Score Cancer Gene Dependency* portal (<https://score.depmap.sanger.ac.uk/>), and also specify the *BRCA2* status of each line. Interestingly, we find that reduced *BRCA2* copy number is more commonly associated with loss of fitness following *PPIA*/*CYPA* loss (**Suppl Table 5**). We also include similar cell line fitness datasets for each of the cancers for whom we demonstrate elevated sensitivity to *CYPAi* (i.e., Neuroblastoma, Multiple Myeloma and CML) (**Suppl Spreadsheet 4**). Fascinatingly, *PPIA*/*CYPA* loss clearly results in loss of fitness in most of these cancer cell lines. Collectively, these new independent comprehensive datasets support our argument that targeting *CYPA* in

Full Revision

select cancer scenarios shows impact in the preclinical setting and may represent an effective new strategy.

The unifying features of the cancers showing elevated sensitivity to CYP*A*i are indeed high genomic instability, denoted by elevated RS and hence a dependency upon replication fork protection machinery. This would be consistent with the observed lethality of our CYP*A*-panel to shBRCA2, siXRCC3 and siRAD51C. The cancers are additionally characterised by aberrantly elevated HRR (i.e. an addiction to/dependency on HRR). This would be consistent with the observed lethality of our CYP*A*-panel to siCtIP, siRAD52, siXRCC3, and siRAD51C. At the Reviewer's request we have reinforced and better clarified this point in the section *Potential rational applications of CYP*A* inhibition in select cancers* and in the *Discussion*.

Reviewer #2.

We thank this reviewer for their positive and supportive comments concerning our work; *"Authors have quite conclusively explored the interaction between NBS1 and cyclophilinA as well as the putative proline residue important for this interaction."* We appreciate the constructive feedback concerning the range of consequences/impacts of CYP*A* impairment and we concur with their contention that *"This manuscript will have broad interest from groups working on genomic stability, immunology as well as cancer therapy."*; a general view also voiced by Reviewer #1.

We do stress that whilst other prolyl isomerases have previously been linked to DNA repair (e.g., most notably the Parvulin family member PIN1), this *is the first time* that CYP*A* has been directly implicated in DNA repair, *and* the first time CYP*A* has been shown to directly interact with a known DNA-R protein (i.e. NBS1).

We believe that the comprehensive CYP*A*-BioID we describe is worthy of report and should serve as a very useful starting point for additional studies concerning CYP*A* biology, which is undoubtedly complex. The interactome will also function as a useful tool in helping dissect the clinically significant wider biological consequences of CYP*A* inhibition. Our interactome findings demonstrate that CYP*A* may influence DNA-R via multiple, and not necessarily mutually exclusive, routes. We do not argue that CYP*A*'s role in DNA-R is exclusively via NBS1/MRN. This is clearly demonstrated by our validation of CYP*A* interactions via co-IP with endogenous CYP*A* with proteins including PCNA, 53BP1, CHAMP1 and ILF2-3 complex (**Fig 5**). These are completely novel observations that furthermore reinforce the validity and efficacy of our experimental approach in leveraging the CYP*A*-BioID to provide new biological insight into this druggable prolyl *cis-trans* isomerase.

Query:

Authors show delayed S-phase transit along with reduced replication speed indicating replication stall. However, authors have not discussed how cyclophilinA might regulate replication (other than hypothesizing regarding altered dynamism of FHA-BRCT). It is conceivable that it could be an indirect effect on cellular metabolism or if authors believe it could be due to direct disruption to core replication machinery or signaling. In this regard, it will be helpful to see if there is shortening of (premature entry) G1 phase and comment on the status of the associated G1/S checkpoint.

RESPONSE:

The reviewer makes a very interesting and astute observation concerning the DNA replication phenotypes we report following CYP A loss and/or inhibition. The bases of these phenotypes are likely multifactorial, and we have revised the associated *Discussion* text to reflect this. Specifically, we highlight the elevated and unresponsive NBS1 and MDC1 foci seen in the CYP A-KO lines (**Fig 9**. i.e., persistent protein-DNA complexes) and dependence upon fork protection factors (XRCC3, RAD51C, BRCA2: **Fig 11**). We also report that a range of DNA replication factors are found in the CYP A-BioID (**Fig 5A**). Untangling the functional significance of these putative interactions would involve further study. Are they direct/indirect interactors? If direct, are they prolyl isomerase substrates or chaperone clients or regulated by liquid-liquid phase separation (LLPS)? Similarly, the CYP A-BioID throws-up an extensive set of RNA binding factors (**Suppl Table 2**), many of whom may conceivably contribute to the replication-transcription fork conflicts/collisions under conditions of CYP A-dysfunction. As this is the first comprehensive report of the cellular impacts of CYP A loss and inhibition, we thought it worth reporting the DNA replication associated phenotypes specifically to demonstrate the pleiotropic impact of loss and inhibition of this particular prolyl isomerase, to underscore its significance/importance. Although we have indeed found cell cycle phase transition impairments in our CYP A-KO and CYP A-R55A cells (for both G1-S and G2-M), these constitute additional studies requiring more thorough molecular-mechanistic characterization. We chose to focus on DNA repair for this first manuscript, as the CYP A-NBS1 interaction was the physical relationship for which we have assembled the most detailed and interconnected datasets, to-date. We do intend to pursue the cell cycle work as it too is derived from our CYP A-BioID (**Suppl Spreadsheet 1**), and we have already validated some of those relevant interactions by CYP A co-IP, but this is very much a work-in-progress. With this manuscript we're endeavoring to tread a fine line by showcasing a wide range of cellular phenotypes resultant from CYP A loss and inhibition, but then also showing a deeper level of characterisation with at least one relevant interactor known to function in a range of DNA-R pathways wherein we've found impairments and dependencies.

Query:

In connection to this, it will also be interesting to see if the ATR/Chk1 signaling axis is intact in CYPA KO cells with or without additional DNA damage compared to WT.

RESPONSE:

At the reviewer's request we include new data showing that HU-induced ATR-dependent CHK1 phosphorylation is normal in CYPA-KO and CYPA-R55A cells, and that ATR does not appear to be spontaneously activated in the absence of replication stress in these cells (**Suppl Fig 1B**).

Query:

Authors show that the P112 residue of NBS1 is important for the binding of cyclophilinA. What is the status of interaction among components of the MRN complex in CYPKO cells and P112G NBS1? Further, what are the authors' thoughts on rescue experiments and whether P112G containing NBS1 to perform resection function.

RESPONSE:

We include new data showing normal expression of MRN components and normal subcellular localisation of NBS1 in the CYPA-KO and CYPA-R55A cells (**Suppl Fig 7A-B**). Regarding the interaction status of P112G, we show that this fails to co-IP endogenous CYPA when transiently expressed in HEK293 cells, in marked contrast to WT-NBS1 (**Fig 8A**). Furthermore, we show that ablation of another FHA Pro residue (P64) does not impair co-IP with endogenous CYPA under similar conditions, suggesting P112G is unique in this regard. Our recombinant protein interaction work demonstrates that CYPA-*Step* directly interacts with a HIS-(FHA-BRCT₁) peptide and that P112G abolishes this interaction (**Fig 8B**). Regarding rescue experiments, we've found that stable overexpression of NBS1 can be neomorphic, resulting in resistance to certain DNA damaging agents, thereby complicating cell-based rescue analyses. We stress that along with our engineered KO and R55A (isomerase-dead) lines we have employed the well-known CYPⁱ Cyclosporin A (CsA) to reproduce several of the DNA-R related phenotypes (e.g., **Fig 1, Fig 3, Fig 6, Fig 10, Fig 11**). To further examine impacts upon resection specifically, a logical approach would be to engineer P112G into a full-length recombinant (baculoviral produced) human MRN complex for *in vitro* kinetic assessment using various labelled DNA substrates. But we think that this specialist and not insignificant undertaking is outside the scope of our report of the extensive *cellular* consequences of CYPA loss and dysfunction and it's potential (pre)clinical significance with regards CYPⁱ repurposing.

Full Revision

Query:

What are the protein levels of MRN, RAD51 etc. in CYPKO cells? It will be important control to delineate the effects of CYP on global transcription and translation vs specific and direct effect on end-resection. Can overexpression of NBS1 rescue the observed resection and focus phenotypes?

RESPONSE:

Basal levels of RAD51 foci/cell are comparable between Scrm and both CYPKO and R55A cells (**Fig 6D**). We also find comparable levels of MRN components between these lines (**Suppl Fig 7A**). Importantly, we observe the pRPA/resection defect following an acute (up to 3hrs) treatment with CsA; conditions unlikely to grossly impair translation to an extent that would result in reduced expression of the relevant DNA-R proteins. Furthermore, microarray based transcriptomic analyses of these isogenic lines did not show evidence of a global impact upon transcription following CYPKO or R55A, nor was there evidence of reduced expression of any genome stability/DNA-R genes. We did not include this negative data so as to maintain the focus on the functional link with DNA repair.

Reviewer #3.

This critically negative review is myopic, unbalanced, self-contradictory and frustratingly mis-represents some of our key findings. The dismissive tone of the text unnecessarily and unprofessionally crosses into the pejorative (*“Either evidence is lacking or experiments were not performed in a convincing way”*). The stark contrast between this review and the summations of Reviewer #1 and Reviewer #2 serve to highlight this hyper-negative approach.

It is very frustrating that this reviewer describes our findings as *“...an interesting story...”*, that *“...the identification of NBS1 as a novel substrate of CYP is significant”*, that the *“..manuscript may provide new insight...”*, and that *“...the role of CYP in DNA repair is fairly well described using its inhibitor or KO cells”*, and yet then concludes by stating *“... the current manuscript suffers lack of evidence to support the main conclusion”*. This is self-contradictory and unbalanced. Again, the contrast with Reviewer #1 and Reviewer #2 in this regard is stark.

Major critical theme no. 1.

Expression of CYP-R55A: *“...vastly different...”*

RESPONSE.

This reviewer dismisses the entirety of the R55A model cell line work based upon the apparent “...*vastly different*...” expression levels of the reconstituted lines. This is an overstatement of the situation and notably not an issue for either Reviewer #1 or Reviewer #2. Nonetheless, we have replaced the original CYPA blot in **Fig 1C** with a clearer and more representative depiction of expression levels between the engineered lines and control. Importantly, the pRPA/resection work, siRAD52 and siXRCC3 dependency work were all corroborated/reproduced using the CYPA PPI inhibitor Cyclosporine A (CsA). The plurality of our complementary approaches showing the influence of CYPA upon DNA-R is minimised and/or ignored by this Reviewer. Although not shown in this study, we find that the R55A cells are selectively sensitive to DNA cross-linker melphalan, in contrast to the CYPA-KO cells. Although we don't yet understand the basis of this observation, this clearly indicates that R55A expression is a valid model in our hands and is not a like-for-like mimic of CYPA-KO simply because of reduced expression. We appreciate the reviewer could not know this.

Major critical theme no. 2.

CYPA-NBS1 work: ***“Another major concern is that the evidence to support that NBS1 is the major substrate of CYPA is lacking since all the experiments were performed with the CYPA mutant or CsA treatment.”***

RESPONSE:

We *do not* claim that NBS1 is “... *the major substrate of CYPA*.”. We *do not* claim that all the DNA-R deficits we have identified are specifically a consequence of impaired NBS1 function. These are misrepresentations of our findings and how we've presented and discussed them. This Reviewer ignores our comprehensive CYPA-BioID, and specifically our discussion pertaining to the DNA-R and Replication factors found therein (section entitled ‘CYPA *Interacting protein partners*’ and **Fig 5A**). We explicitly discuss the fact that “*A recurring theme amongst these CYPA interactors is that all are involved in end-resection*” whilst also demonstrating CYPA co-IP with 53BP1, CHAMP1 and ILF2-3 (**Fig 5C-E**). In the ‘*Discussion*’ section we describe a “*homesostatic role for CYPA in genome stability*”, including possible contributions to controlling LLPS of well-known DNA-R factors and the fact that several mitotic, kinetochore, centrosomal and spindle proteins are found in the CYPA-BioID.

Major critical theme no. 3.

A major repeated criticism levelled by this reviewer as a basis for dismissing the entirety our findings is that we have failed to demonstrate that the catalytic activity of CYPA is required for DSB repair.

- ***Their conclusion should be supported by additional key experiments to prove that the catalytic activity of CYPA is indeed required for DSB repair...***
- ***Another major concern is that the evidence to support that NBS1 is the major substrate of CYPA is lacking since all the experiments were performed with the CYPA mutant or CsA treatment.***
- ***One major weakness of this study is that it focuses on characterizing the interaction between CYPA and NBS1, then jumps into a conclusion that the catalytic activity of CYPA is required for DSB repair based on its direct interaction with NBS1***

RESPONSE:

As this criticism is repeated, the impression created, and no doubt intended, is that the manuscript is irreparably flawed (“...major weakness...”). This is an over-simplification and a misdirection. It’s notable that this critique isn’t raised in such a manner by either Reviewer #1 or Reviewer #2. This is likely because any modest inferences we made concerning the possible role of CYPA catalytic isomerase activity were based on a combination of differing but complementary approaches. Firstly, we routinely used the p.R55A engineered CYPA variant, although this Reviewer regards our use of this as invalid. This longstanding peptidyl prolyl isomerase (PPI)-dead mutant model has frequently been employed to invoke the catalytic function of CYPA. The mutant was originally proposed and characterized as catalytically-dead using the *in vitro* chymotrypsin-coupled prolyl isomerase assay using *N*-succinyl-AAPF-*p*-nitroanilide as a substrate as far back as 1992 (Zydowsky, L.D., et al., *Active site mutants of human cyclophilin A separate peptidyl-prolyl isomerase activity from cyclosporin A binding and calcineurin inhibition*. Protein Science, 1992. 1(9): p.1092-1099). In addition, we routinely use Cyclosporin A (CsA), the longstanding clinically relevant CYPA PPI inhibitor, and we also use a different and more potent CYPA PPI inhibitor, namely NIM811 (*N*-methyl-4-isoleucine-cyclosporine) for the DR-GFP reporter assays of individual DNA-R pathway function (i.e.’ NHEJ, HRR and SSA).

With regards to our findings concerning CYPA-NBS1 interaction, in the *Discussion* section we clearly state that mechanistic analyses of prolyl isomerase on the dynamism of NBS1 FHA-BRCT would require specialist approaches outside the scope of this manuscript, as the manuscript is firmly within the realm of cellular biology. This is ignored by this Reviewer. Specifically, we state that “*A regulated cis-trans isomerisation of the E111-P112 peptide bond could conceivably dynamically alter the relative positioning of the FHA domain with the tandem BRCTs of NBS1 (Fig 7C-D). This may then impact on these domains’ abilities to dynamically interact with their respective phospho-threonine (for FHA) and phospho-serine (BRCT) containing targets, consequently likely shaping/impacting NBS1 recruitment dynamics and/or plasticity of its interactome [120-122]. Demonstrating this hypothesis would require additional structural analysis using techniques such as 2D-NMR which is outside the scope of this manuscript.*”

Minor comments: 1.

Fig. 1E; is the survival between KO and R55A statistically significant? If so, do the authors have an explanation? Why is the reconstitution of R55A more toxic than KO alone?

RESPONSE:

Yes, R55A is slightly more sensitive compared to KO for this endpoint. The irony that this observation runs contrary to the Reviewer's dismissal of the R55A model line is not lost on us (**Major critical theme no. 1**). As is well-known for PARP1, inhibition is not equivalent to absence. A possible speculative explanation is that the R55A isomerase-dead could have additional dominant impacts compared to the KO situation. Nonetheless, we suspect this Reviewer would object to such speculation in the absence of ever more data.

Minor comments: 2.

In Fig. 3D, the NHEJ activity of CsA- or NIM811-treated cells is significantly downregulated in comparison to control, which raises the possibility of the pleiotropic effect of CYPA inhibition. The authors should discuss this issue.

RESPONSE:

Not necessarily indicative of a pleiotropic effect if one accepts that absence of a protein is not always biologically equivalent to the presence of an inhibited version the same protein. Of note, we do see somewhat reduced NHEJ following siCYPA (**Fig 3A**), something not mentioned by this Reviewer. Furthermore, we explicitly discuss and show interaction between CYPA and 53BP1, CHAMP1 and ILF2-3 complex, all players in NHEJ and in the intricate balance between NHEJ and resection-mediated recombination directed repair pathways.

Minor comments: 3.

In Figure 8A, since the expressions of Flag-NBS1 WT, P112G, and P64G are very different, the conclusion that the isomerization of CYPA is essential for NBS1 cannot be supported. Given the variation of input levels, it appears that the P64G mutation actually enhances the interaction with endogenous CYPA. Is this reproducible? This co-IP result may need to be quantified from independent sets for statistical analysis.

RESPONSE:

We do not claim that "...isomerization of CYPA is essential for NBS1...". **Fig 8A** data is derived from a transient transfection. Whilst there is some variation in expression, we do not make any precise quantitative conclusions from these co-IPs. Nonetheless, FLAG-NBS1-P112G clearly interacts less with endogenous CYPA in this system. Importantly, and ignored by this

Full Revision

Reviewer, the associated recombinant protein work shown in **Fig 8B** clearly confirms that NBS1-P112G is profoundly compromised in its ability to interact with CYP A.

Minor comments: 4.

A defect in DSB repair generally hypersensitizes cells to DNA replication stress, including HU. In this regard, resistance of the CYP A KO (or R55A cells) to HU is interesting, but it may be due to the nonspecific effect of the CYP A loss in multiple DNA damage signaling and repair processes. Alternatively, cell cycle may be affected nonspecifically, rendering cells resistant to replication-associated genotoxic stress. This needs to be addressed further. Analysis of overall cell cycle profile may be required.

RESPONSE:

Resistance to HU is likely multifactorial and cell cycle transition kinetics may be relevant here. That is why we linked the DNA replications phenotypes to this discussion in the section entitled “*Impaired CYP A function reveals novel genetic dependencies/vulnerabilities*”. A comprehensive analysis of cell cycle profile and phase transits is outside the scope of the current manuscript (see response to **Reviewer #2**).

Impaired HU-induced pRPA has been linked to HU-resistance via NBS1 previously: Manthey, K.C., et al., *NBS1 mediates ATR-dependent RPA hyperphosphorylation following replication-fork stall and collapse*. J Cell Sci, 2007. **120**(Pt 23): p. 4221-9.

Minor comments: 5.

Text not to mention Abstract is too dense. The manuscript will benefit a lot from extensive editing and rearrangement of figures to make the story more succinct for journal submission.

RESPONSE:

The Reviewer’s view concerning a lack of succinctness is not shared by Reviewer #1 and Reviewer #2. We have endeavored to draft a concise and accessible manuscript, the main body of which comes in at just over 23x sides of A4 (including Materials & Methods). Considering we guide the reader through 12x multipart figures, 5x supplementary tables and 8x supplementary figure, we believe we have achieved succinctness. Nonetheless, we will of course take direction from the appropriate journal editorial team regarding house style and format.

9th Feb 2024

RE: Manuscript EMBOR-2024-58928V1, A novel role for the peptidyl-prolyl cis-trans isomerase Cyclophilin A in DNA-repair following replication fork stalling via the MRE11-RAD50-NBS1 complex.

Dear Mark,

Thank you for the submission of your revised ms to EMBO reports. I have now read and discussed it with the EMBO reports team, including our chief editor Bernd Pulverer.

We certainly appreciate that your study reveals a role for CYP A in DNA repair/end resection, and that you identify a number of CYP A interacting partners, including NSB1. We also think that the title of your ms is interesting, that CYP A has a role in DNA-repair following replication fork stalling via the MRE11-RAD50-NBS1 complex. If you can provide stronger data to support this statement, we would be interested in your ms for publication by EMBO reports. Given that you generated the NBS1-P112G mutant, that does not interact with CYP A, we would like to know whether cells expressing this mutant have DNA repair or resection phenotypes? In the absence of stronger data that support a role of CYP A in DNA repair through NBS1, we think that the ms would be better suited for our open access sister journal Life Science Alliance.

You can either re-submit a strengthened ms to EMBO reports, or use the link below to transfer your ms to Life Science Alliance. Upon transfer to LSA, you can discuss with Eric Sawey (e.sawey@life-science-alliance.org) the exact revision requirements for LSA.

As a service to authors, EMBO provides authors with the possibility to transfer a manuscript that one journal cannot offer to publish to another EMBO publication. The full manuscript and if applicable, reviewers reports are automatically sent to the receiving journal to allow for fast handling and a prompt decision on your manuscript. For more details of this service, and to transfer your manuscript to another EMBO title please click on Link Not Available

=====

Rev_Com_number: RC-2023-02068

New_manu_number: EMBOR-2024-58928V1

Corr_author: O'Driscoll

Title: A novel role for the peptidyl-prolyl cis-trans isomerase Cyclophilin A in DNA-repair following replication fork stalling via the MRE11-RAD50-NBS1 complex.

4th Apr 2024

Manuscript number: RC-2023-02068R and **EMBOR-2024-58928V1**
Corresponding author: Prof. Mark O'Driscoll

Re: Submission of revised manuscript entitled “A novel role for the peptidyl-prolyl *cis-trans* isomerase Cyclophilin A in DNA-repair following replication fork stalling via the MRE11-RAD50-NBS1 complex”.

Bedir M et al. BioRxiv 2023-546694v1. <https://doi.org/10.1101/2023.06.27.546694>

FAO: Dr. Esther Schnapp, Senior Editor, *EMBO Reports*.

Dear Esther,

I previously transferred a peer reviewed (and post-review revised) version of this manuscript to *EMBO Reports* via the *Review Commons* platform (02-Feb-2024). It was assigned the manuscript no **EMBOR-2024-58928V1**. Following consultation with *Review Commons* and yourself, I'm submitting this as a 'new submission', as instructed.

Following review by you and your editorial team, you requested we attempt to generate additional new data concerning DNA repair function for NBS1-P112G, a variant that fails to bind the peptidyl-prolyl *cis-trans* isomerase Cyclophilin A (CYPA):

“...We certainly appreciate that your study reveals a role for CYPA in DNA repair/end resection, and that you identify a number of CYPA interacting partners, including NSB1. We also think that the title of your ms is interesting, that CYPA has a role in DNA-repair following replication fork stalling via the MRE11-RAD50-NBS1 complex. If you can provide stronger data to support this statement, we would be interested in your ms for publication by EMBO reports. Given that you generated the NBS1-P112G mutant, that does not interact with CYPA, we would like to know whether cells expressing this mutant have DNA repair or resection phenotypes?...”

I am happy to report that we have successfully generated said data and have now integrated this important addition into our CYPA manuscript. Therefore, we are delighted to offer this fully revised draft for consideration as the first multifaceted and comprehensive demonstration of a novel link between the peptidyl prolyl *cis-trans* isomerase CYPA and the DNA repair/genome stability network.

Mark O'Driscoll.
Professor of Human Molecular Genetics.
Human DNA Damage Response Disorders Group.

Genome Damage & Stability Centre
University of Sussex
Falmer, Brighton, BN1 9RQ, United Kingdom
Ph. 0044(0)1273 877 515
m.o-driscoll@sussex.ac.uk
<http://www.sussex.ac.uk/lifesci/odriscolllab/>

New DNA repair data concerning NBS1-P112G.

Using the Nijmegen Breakage Syndrome patient-derived SV40-transformed skin fibroblast NBS-ILB1 as a host, we expressed wild-type (WT) and the NBS1-P112G variant, then used these lines to assess HU-induced NBS1 and RAD51 foci formation. NBS-ILB1 is homozygous for the *NBN* founder mutation [(657del5), p.Lys219Asnfs*16 (c.657_661delACAAA)], and does not express full length NBS1. When using an N-ter epitope-directed anti-NBS1 antibody, no protein signal whatsoever is detected in these cells, making this system an extremely 'clean' and sensitive route to investigate NBS1 foci formation from ectopically expressed sources. We find significant impairment in foci formation for both NBS1 and RAD51 following expression of NBS1-P112G, in contrast to NBS1-WT. These data demonstrate clearly impaired engagement of MRN complex mediated resection-driven homologous recombination repair following DNA replication fork stalling for NBS1-P112G, a variant that fails to sustain stable direct binding to CYP A. These data are included as a revised **Fig 9(C-D)**.

Key novel summary findings detailed in our manuscript.

In this paper we describe a CYP A-BioID generated interactome, identifying a plethora of novel putative CYP A interactors. We validate several interactions with well-characterised DNA-R proteins and show that CYP A can directly interact with NBS1. We document a range of DNA-R deficits caused by loss and/or inhibition of CYP A and identify a series of CYP A dependencies/vulnerabilities within the genome stability network. Collectively, these represent completely new biological insights into the consequences of impaired CYP A function. Finally, we demonstrate the targeted application of CYP A inhibition can be effective in a set of cancers sharing a genomic profile of elevated replication stress and homologous recombination dependency/addiction. This is significant, as several clinically validated CYP A inhibitors already exist (e.g., Alisporivir, NIM811, SCY-635, Rencofilstat). Whilst these non-immunosuppressive Cyclosporin A analogues are currently being trialled as anti-virals (e.g., against pan-genotypic hepatitis, HIV, SARS-CoV2) and antifibrotics (e.g., for non-alcoholic steatohepatitis), we provide evidence that strongly argues that their repurposing/repositioning against select cancers in a precision fashion should be considered.

We believe our findings are of interest to a wide readership, a point reflected by Reviewer #1's comment: "*It will spark interest from molecular biologists to clinicians and pharmaceutical researchers*". Considering we present a comprehensive CYP A interactome thereby suggesting a wealth of additional research avenues worth pursuing, we think the wider impact and future citation potential of this original paper are extremely high. As well as also claiming primacy for a functional interaction between CYP A and the DNA repair machinery, our manuscript strongly proposes a clear evidential pathway for investigating the potential of repurposing already existing clinically validated CYP A inhibitors. Pharmacological CYP A inhibition is an active area of interest by international Pharma as demonstrated by companies such as Cypralis Ltd (UK), Hepion Pharmaceuticals (US) and DebioPharm SA (CH). In fact, Hepion have already pivoted towards investigating their bespoke CYP A inhibitor Rencofilstat for hepatocellular carcinoma (<https://hepionpharma.com/newsroom/>).

All the best

Mark O'Driscoll
(on behalf of the authors).

Dear Mark,

Thank you for the submission of your revised manuscript to EMBO reports. We have now received the enclosed reports from 2 referees. Previous referee 3 decided to withdraw from the review process, and I therefore contacted another expert, referee 4 below.

As you will see, referee 4 agrees with some of the points raised by referee 3 and referee 2 also agrees in the cross-comments that a few more analyses could have been done. I would like to know which of these points you are willing to address? It would be good if you could get back to me by tomorrow as I will be out of the office for the following 2 weeks.

A few editorial requests will also need to be addressed:

- Please submit your final ms also as a word file without figures.
- All main and EV figures need to be uploaded as individual files.
- The number of keywords needs to be reduced to 5, "and phrases" needs to be deleted.
- A Data Availability Section (DAS) is missing at the end of the materials section. If no data have been deposited in public databases, this fact needs to be mentioned in the DAS.
- Please rename the conflict of interest subheading to "Disclosure Statement and Competing Interests"
- The REFERENCE FORMAT needs to be alphabetical, et al needs to be used after 10 author names; DOIs should only be used for preprints and datasets that have not been published yet. The EMBO reports reference style is also in EndNote.
- DATA NOT SHOWN page 54 needs to be removed as per journal policy.
- Please co-submit with your final ms a completed author checklist that can be found here: .
The completed list will also be part of our transparent peer-review process file (RPF).
- The excel file needs to be called Dataset EV1, please also correct the callouts in the ms file accordingly.
- The suppl. tables and figures could all be placed in a PDF Appendix file as "Appendix Table S1", etc. "Appendix Figure S1", etc, their legends included. It appears that there are 2 sets of Supplementary Tables 1-4, this needs to be corrected, renamed, etc.; Supplementary Materials & Methods should also be part of the Appendix and need to be renamed to "Appendix Materials and Methods". The Appendix file needs to have a Table of Content with page numbers on the first page. All callouts in the ms text need to be updated accordingly.
- We can also accommodate up to 6 EV (Expanded View) figures which would need to be uploaded as individual Figure files and their legends would need to go into the ms after the main legends. The callouts in the ms for all these figures and tables need to be updated accordingly. You can find more information about our file types in our guide to authors online:
<https://www.embopress.org/page/journal/14693178/authorguide#manuscriptpreparation>
- The manuscript sections should be in the following order: Title page - Abstract & Keywords - Introduction - Results - Discussion - Methods - Data Availability - Acknowledgments - Disclosure Statement & Competing Interests - References - Figure Legends - Tables with legends - Expanded View Figure Legends.
- Please also address the following comments by our data editors:
 1. Please note that the individual figure legends for supplementary figures 8a-g is not provided in the manuscript. This needs to be rectified.
 2. Please note that the exact p values are not provided in the legends of figures 1b, e; 2b-c; 3a-d; 6c; 9a-b, d-e; 10a-b; 11c-d.
 3. Please indicate the statistical test used for data analysis in the legend of figure 4b.
 4. Please note that the box plots need to be defined in terms of minima, maxima, centre, bounds of box and whiskers, and percentile in the legend of figure 2c.
 5. Please note that the box plots need to be defined in terms of minima, maxima, bounds of box and whiskers, and percentile in the legends of figures 6c; 9a-b, d-e.
 6. Please note that information related to n is missing in the legends of figures 6c; 11e; 12b, d, f-g.
 7. Please note that the error bars are not defined in the legends of figures 11e; 12b, d, f-g.
 8. Please note that scale bar and its definition are missing for figure 1a.

EMBO press papers are accompanied online by A) a short (1-2 sentences) summary of the findings and their significance, B) 2-3 bullet points highlighting key results and C) a synopsis image that is exactly 550 pixels wide and 200-600 pixels high (the height is variable). The synopsis image should provide a sketch of the major findings, like a graphical abstract. Please note that text needs to be readable at the final size. Please send us this information along with the final manuscript.

I look forward to seeing a final version of your manuscript when it is ready.

Referee #1:

The investigators have adequately addressed my concerns. The paper should be accepted.

Referee #2:

The authors highlight the role of CYPA in regulating end-resection machinery. Further, they describe a novel direct interaction between NBS1 and CYPA, the disruption of which, interferes with the function of the MRN complex. The authors also explored genetic vulnerabilities associated with CYPA loss. Overall, I feel the authors have done thorough work towards the main conclusions of the manuscript. The study also opens new questions that can be pursued by authors or other researchers in the field.

Referee #4:

In my opinion, the authors might evaluate to reduce the initial part of the study to a minimum and even put the results shown in the first 3 figures after the "biochemical" sections when they report on the significance of the interaction and, as such, of the CYPA-mediated modification of NBS1. This is just my suggestion.

I do not agree with the authors' negative feedback on the request for even a simple analysis of cell cycle in CYPA KO or R55A cells (or CYP*A*i-treated ones).

It seems to me reasonable to address some of the comments raised by reviewer 3 and 1 (check cell cycle overall, one or two easy cell biology experiments to show that R55A is phenotypic beyond its reduced expression level, focus on HR ditching the experiments on fork processing and discussing about the limitation of the study, which is focused on the HR-related NBS1 function) but that the work is worth to be published in EMBO Reports.

Cross-comments from referee 2:

I looked at reviewer 3's concerns and Figure 9.

Personally, contrary to the author's view, I found comments/concerns from reviewer 3 were genuine and anyone reading this manuscript will legitimately have those questions.

The authors should have taken criticism constructively and tried addressing it calmly, omitting all sarcasm in response to the reviewer rather than taking it negatively.

Now, a question regarding the mutant level being low (WT reconstitution being missing) and pleiotropic effects have been answered by providing explanations. Authors surely could have done better by experimentally reconstituting WT as reviewer 3 suggested at least for Figure 1. Here, I would also like to mention that most of the effects authors see are probably indirect or result from one or multiple targets. The authors are careful in mentioning this in the manuscript as well.

Overall, the authors do show the effects of cyclophilin A in end-resection (through multiple mechanisms/targets both direct or indirect perhaps) and that it interacts with NBS1 and through proline 112 on NBS1.

I based my decision on the fact that the authors do have data supporting these major findings as it is the central conclusion of their manuscript.

The physiological relevance of this interaction, the direct role CycA in DNA resection, the efficacy of its inhibitor in BRCA loss

background, etc at this stage is preliminary and requires more thorough examination but probably beyond the scope of this manuscript.

Rev_Com_number: RC-2023-02068

New_manu_number: EMBOR-2024-58928V2-Q

Corr_author: O'Driscoll

Title: A novel role for Cyclophilin A in DNA-repair via the MRE11-RAD50-NBS1 complex.

17th May 2024

Manuscript number: EMBOR-2024-58928V2-Q
& RC-2023-02068 [REV]

Corresponding author: Prof. Mark O'Driscoll

Re: Response to Referees following Revision.

"A novel role for the peptidyl-prolyl cis-trans isomerase Cyclophilin A in DNA-repair following replication fork stalling via the MRE11-RAD50-NBS1 complex".

Bedir M et al. BioRxiv 2023-546694v1. <https://doi.org/10.1101/2023.06.27.546694>

FAO: Dr. Esther Schnapp, Senior Editor, *EMBO Reports*.

Dear Esther,

It is highly regrettable that the original Referee #3 has refused to handle the revised manuscript, especially considering the effort made in addressing their specific concerns which amounted to 4.5 Ax pages in the **Review Commons Full Revision Document**. I understand how this situation has consequently delayed consideration of the revised manuscript. I believe that was their intention.

A point-by-point response to each referee is as follows:

Referee #1.

The investigators have adequately addressed my concerns. The paper should be accepted.

RESPONSE.

Thanks for the support and affirmative judgement.

Mark O'Driscoll.
Professor of Human Molecular Genetics.
Human DNA Damage Response Disorders Group.

Genome Damage & Stability Centre
University of Sussex
Falmer, Brighton, BN1 9RQ, United Kingdom
Ph. 0044(0)1273 877 515
m.o-driscoll@sussex.ac.uk
<http://www.sussex.ac.uk/lifesci/odriscolllab/>

Referee #2.

The authors highlight the role of CYPA in regulating end-resection machinery. Further, they describe a novel direct interaction between NBS1 and CYPA, the disruption of which, interferes with the function of the MRN complex. The authors also explored genetic vulnerabilities associated with CYPA loss.

Overall, I feel the authors have done thorough work towards the main conclusions of the manuscript. The study also opens new questions that can be pursued by authors or other researchers in the field.

RESPONSE.

We appreciate the positive summation and especially the description of our work as “thorough”. Indeed, we believe that our interactome findings, and not simply the DNA-R aspect of our work, will help seed future avenues for researchers to pursue.

Referee #4.

In my opinion, the authors might evaluate to reduce the initial part of the study to a minimum and even put the results shown in the first 3 figures after the "biochemical" sections when they report on the significance of the interaction and, as such, of the CYPA-mediated modification of NBS1. This is just my suggestion.

We appreciate the thoughtful suggestion, but feel the narrative works better when directly following-on from the original report of the sensitivity of *LIG4*^{-/-} human pre-B lymphocytes to killing by Cyclosporin A (CsA), and that CsA induces DNA double strand breaks only in cells that have traversed S phase (O’Driscoll M et al *Bone Marrow Transplant*, 2008, *pmid*: 18278071). This manuscript begins by proving that loss or catalytic inhibition of Cyclophilin A (CYPA), the principal physiological protein target of CsA, likely underlies those original observations. We also believe that presenting the story in this fashion will be of appeal to the clinical and industrial (i.e. Cyclophilin inhibitor) communities. This is a large and data-dense manuscript befitting the range of priority observations and claims we report.

I do not agree with the authors' negative feedback on the request for even a simple analysis of cell cycle in CYPA KO or R55A cells (or CYPAl-treated ones).

We are confused and disappointed that our detailed and reasoned response to this point, is described by this reviewer as “*negative feedback*”. That was not our intention (see further comments below concerning cross-comments from Referee #2).

We believe we have dealt sufficiently and comprehensively with this issue in our original response to Referee #2 (**Review Commons Full Revision Document, p7**) and Referee #3 (**Review Commons Full Revision Document, p13, Minor comments:4**). We have modified the *Discussion* text to address the likely multifactorial origin of the DNA replication deficits we described in Fig 2. We do not dispute the likely pleiotropic impacts of CYPA loss/inhibition on cell cycle dynamics. Indeed, we believe this is precisely why these impacts such as these are interesting and physiologically relevant in the context of clinical inhibition of CYPA. Specifically, we highlight the elevated and

unresponsive NBS1 and MDC1 foci seen in the CYP A-KO lines (**Fig 9**. i.e., persistent protein-DNA complexes) and dependence upon fork protection factors (XRCC3, RAD51C, BRCA2: **Fig 11**). We also report that a range of DNA replication factors are found in the CYP A-BioID (**Fig 5A**). Untangling the functional significance of these putative interactions would most definitely involve further study. At present, this manuscript is composed of 50x individual data panels (for Figs 1-12), 30x data panels for Suppl Figs 1-8, 4x Suppl Tables and 4s Suppl data Spreadsheets. We do not think a wider cell cycle analysis will add any depth of mechanistic understanding and so respectfully disagree with the reviewer on this point.

It seems to me reasonable to address some of the comments raised by reviewer 3 and 1 (check cell cycle overall, one or two easy cell biology experiments to show that R55A is phenotypic beyond its reduced expression level, focus on HR ditching the experiments on fork processing and discussing about the limitation of the study, which is focused on the HR-related NBS1 function) but that the work is worth to be published in EMBO Reports.

RESPONSE.

We have included new R55A expression data as Suppl Fig 1A and although somewhat reduced, it's important to note that R55A is not a major component of our paper, and key cellular data concerning DNA-R or cell sensitivities are reproduced with the clinical inhibitor of CYP A, Cyclosporin A (CsA). Our feeling was that the original Referee 3 over-emphasised the point concerning R55A levels to help justify the overly negative tone of their review of the original manuscript. We made that point openly in response to that individual in full expectation that they were free to counter upon re-review. They have decided not to have anything more to do with this paper. We are concerned at the data density of the current manuscript and believe the inclusion of additional cell cycle data, which by its nature will be descriptive, will further imbalance the narrative and not increase depth of mechanistic understanding.

Importantly, we have found cell cycle phase transition impairments in our CYP A-KO and CYP A-R55A cells (for both G1-S and G2-M), but these descriptive observations require a more thorough molecular-mechanistic characterisation. We chose to focus on DNA repair here, as the CYP A-NBS1 interaction was the physical relationship for which we have assembled the most detailed and interconnected datasets. Cell cycle constituents are present in our CYP A-BioID and we highlight this (**Suppl Spreadsheet 1**). With this manuscript we've endeavoured to tread a fine line by showcasing a wide range of cellular phenotypes resultant from CYP A loss and inhibition, but then also showing a deeper level of characterisation with at least one relevant interactor known to function in a range of DNA-R pathways.

Cross-comments from referee 2:

I looked at reviewer 3's concerns and Figure 9.

Personally, contrary to the author's view, I found comments/concerns from reviewer 3 were genuine and anyone reading this manuscript will legitimately have those questions.

The authors should have taken criticism constructively and tried addressing it calmly, omitting all sarcasm in response to the reviewer rather than taking it negatively.

Whilst Referee #2 is entitled to their opinion concerning the review drafted by Referee #3, it is important to note that Referee #3 have removed themselves from the process, apparently in a fit of pique. That is inconsistent with being genuine.

It is of concern to us that we have circled back, again, to this reviewer. I have discussed Referee #3's response with senior colleagues here and can confirm that they do not share their opinion. It is important to remember that they claimed we had not performed experimental work properly (i.e., incompetence), nor adequately drafted the manuscript (i.e., unskilled, inexperienced amateurs). A reasonable interpretation is that that was unnecessarily personal. It is certainly not "constructive". They did this behind the protection of anonymity and in the knowledge that their review would be published on *BioRxiv*, as this was a submission via *Review Commons*. As a duty to my team, I was compelled to forcibly counter their criticisms and make no apology. Such negativity has real-world consequences for the employment prospects particularly for the more junior members of the team. I had to ensure our collective response was clear, called out instances where we felt there were intentional mis-directions and for our response to sit alongside that original review.

"Sarcasm" was not a feature of our original response, and we strongly reject that claim. Our response was robust and calibrated to counter the highly damaging and pejorative accusations levelled concerning our fundamental competence.

Now, a question regarding the mutant level being low (WT reconstitution being missing) and pleiotropic effects have been answered by providing explanations. Authors surely could have done better by experimentally reconstituting WT as reviewer 3 suggested at least for Figure 1. Here, I would also like to mention that most of the effects authors see are probably indirect or result from one or multiple targets. The authors are careful in mentioning this in the manuscript as well.

We have dealt with this in the revised draft (as highlighted by this referee), and in the relevant comments above.

Overall, the authors do show the effects of cyclophilin A in end-resection (through multiple mechanisms/targets both direct or indirect perhaps) and that it interacts with NBS1 and through proline 112 on NBS1.

I based my decision on the fact that the authors do have data supporting these major findings as it is the central conclusion of their manuscript.

The physiological relevance of this interaction, the direct role CycA in DNA resection, the efficacy of its inhibitor in BRCA loss background, etc at this stage is preliminary and requires more thorough examination but probably beyond the scope of this manuscript.

We appreciate the reviewer's supportive judgement that we have provided data to support the major findings. We concur that there is much more to uncover concerning mechanisms and potential applications and believe this manuscript will serve as an important starting point for others.

Best regards

Mark O'Driscoll
(on behalf of the authors).

28th May 2024

Manuscript number: EMBOR-2024-58928V2-Q
& RC-2023-02068 [REV]

Corresponding author: Prof. Mark O'Driscoll

Re: Submission of Revision R2.

"A novel role for the peptidyl-prolyl cis-trans isomerase Cyclophilin A in DNA-repair following replication fork stalling via the MRE11-RAD50-NBS1 complex".

Bedir M et al. BioRxiv 2023-546694v1. <https://doi.org/10.1101/2023.06.27.546694>

FAO: Dr. Esther Schnapp, Senior Editor, *EMBO Reports*.

Dear Esther,

Please find enclosed a revised draft of the above manuscript, along with Figures, Appendix, EV-datasets and the requested Source Data files. The manuscript and supplementary (Appendix) information have been reconfigured, amended, and rearranged into the format requested (*'editorial requests'*). All necessary figure and table call-outs have also been amended. Please also find enclosed a completed *Author Checklist* and *Source Data checklist*. We have assembled all of the requested Source Data files into Fig-specific Zip folders, as requested. We have also included a brief synopsis and associated graphic, as requested. For reference, please also find at the end of this letter a copy of our rebuttal to the points raised by the referees upon re-review, that was returned via email on 17th May 2024.

In response to the specific comments raised by the journal data editors:

1. Please note that the individual figure legends for supplementary figures 8a-g is not provided in the manuscript. This needs to be rectified.

Appendix Fig 8 legend change as per instruction.

2. Please note that the exact p values are not provided in the legends of figures 1b, e; 2b-c; 3a-d; 6c; 9a-b, d-e; 10a-b; 11c-d.

Mark O'Driscoll.
Professor of Human Molecular Genetics.
Human DNA Damage Response Disorders Group.

Genome Damage & Stability Centre
University of Sussex
Falmer, Brighton, BN1 9RQ, United Kingdom
Ph. 0044(0)1273 877 515
m.o-driscoll@sussex.ac.uk
<http://www.sussex.ac.uk/lifesci/odriscolllab/>

Exact *p* values now included in all relevant legends and are also provided in the respective Source Data files.

3. Please indicate the statistical test used for data analysis in the legend of figure 4b.

Fischer's Exact Test. The legend has been amended to include this information.

4. Please note that the box plots need to be defined in terms of minima, maxima, centre, bounds of box and whiskers, and percentile in the legend of figure 2c.

All data blot parameters are now included as 'Box Plot Values' summary document in for each relevant figure in their specific Source Data file.

5. Please note that the box plots need to be defined in terms of minima, maxima, bounds of box and whiskers, and percentile in the legends of figures 6c; 9a-b, d-e.

All data blot parameters are now included as 'Box Plot Values' summary document in for each relevant figure in their specific Source Data file.

*6. Please note that information related to *n* is missing in the legends of figures 6c; 11e; 12b, d, f-g.*

Information now included in the revised figure legends.

7. Please note that the error bars are not defined in the legends of figures 11e; 12b, d, f-g.

Information now included in the revised figure legends.

8. Please note that scale bar and its definition are missing for figure 1a.

Scale bar (10 μ m) now included in the revised Fig 1A.

Principal novel summary findings of our manuscript.

We describe a comprehensive CYPA-BioID generated interactome, identifying a plethora of novel putative CYPA interactors. We validate several interactions with well-characterised DNA-R proteins and show that CYPA can directly interact with NBS1. We document a range of DNA-R deficits caused by loss and/or inhibition of CYPA and identify a series of CYPA dependencies/vulnerabilities within the genome stability network. Collectively, these represent completely new biological insights into the consequences of impaired CYPA function. Finally, we demonstrate the targeted application of CYPA inhibition can be effective in a set of cancers sharing a genomic profile of elevated replication stress and homologous recombination dependency/addiction. This is significant, as several clinically validated CYPA inhibitors already exist (e.g., Alisporivir/Debio-025, NIM811, SCY-635, Rencofilstat/CRV431). Whilst these non-immunosuppressive Cyclosporin A analogues are currently being trialled as anti-virals (e.g., against pan-genotypic hepatitis, HIV, SARS-CoV2) and antifibrotics (e.g., for non-alcoholic steatohepatitis), we provide evidence that strongly argues that their repurposing/repositioning against select cancers in a precision fashion should be considered.

Impact and appeal of our findings.

We believe our findings are of interest to a wide readership, a point reflected by Reviewer #1's comment: "*It will spark interest from molecular biologists to clinicians and pharmaceutical researchers*". Considering we present a comprehensive CYPA interactome thereby suggesting a wealth of additional research avenues worth pursuing, we think the wider impact and future citation potential of this original paper are extremely high.

As well as also claiming primacy for a functional interaction between CYPA and the DNA repair machinery, our manuscript strongly proposes a clear evidential pathway for investigating the potential of repurposing already existing clinically validated CYPA inhibitors. Pharmacological CYPA inhibition is an active area of interest by international Pharma as demonstrated by companies such as Cypralis Ltd (UK), Hepion Pharmaceuticals (US) and DebioPharm SA (CH). In fact, The Nasdaq listed Cyclophilin inhibitor company Hepion Pharmaceuticals has already pivoted towards investigating their bespoke CYPA inhibitor Rencofilstat for hepatocellular carcinoma (<https://hepionpharma.com/newsroom/>).

If there is anything else you need let me know.

Best regards

Mark O'Driscoll
(on behalf of the authors).

Prof. Mark O'Driscoll
Sussex, University of
Genome Damage and Stability Unit
University of Sussex
Falmer
Brighton, UK-East Sussex BN1 9RQ BN1 9RQ
United Kingdom

Dear Prof. O'Driscoll,

I am very pleased to accept your manuscript for publication in the next available issue of EMBO reports. Thank you for your contribution to our journal.

Yours sincerely,

Rev_Com_number: RC-2023-02068
New_manu_number: EMBOR-2024-58928V3
Corr_author: O'Driscoll
Title: A novel role for Cyclophilin A in DNA-repair via the MRE11-RAD50-NBS1 complex.